# Articulate your NeRF: Unsupervised articulated object modeling via conditional view synthesis

**Jianning Deng    Kartic Subr    Hakan Bilen**
University of Edinburgh
https://github.com/VICO-UoE/ArticulateYourNerf/

## Abstract

We propose a novel unsupervised method to learn pose and part-segmentation of articulated objects with rigid parts. Given two observations of an object in different articulation states, our method learns the geometry and appearance of object parts by fitting an implicit model on the first observation and renders the latter observation by distilling the part segmentation and articulation. Additionally, to tackle the challenging joint optimization of part segmentation and articulation, we propose a voxel grid based initialization strategy and a decoupled optimization procedure. Compared to the prior unsupervised work, our model obtains significantly better performance, generalizes to objects with arbitrary number of parts while it can be efficiently learned from few views only for the latter observation.

## 1   Introduction

Articulated objects, composed of multiple rigid parts connected by joints allowing rotational or translational motion, such as doors, cupboards and spectacles are ubiquitous in our daily lives. Automatically understanding the shape, structure and motion of these objects is crucial for numerous applications in robotic manipulation [12, 43] and character animation [1, 32]. Many works [10, 30, 34] that focused on this problem use groundtruth 3D shape, articulation information, and/or part segmentation to learn articulated object models but acquiring accurate 3D observations and manual annotations is typically complex and too expensive for building real large-scale datasets.

In this paper, we introduce a novel unsupervised technique that learns part segmentation and articulation (*i.e.*, axis of movement, and translation/rotation of each movable part) from two sets of observations *without* requiring groundtruth shape, part segmentation or articulation. Two sets contain images of the same object from multiple viewpoints in two different articulation states. Our key idea is that articulation changes only the poses of the object parts but not their geometry or texture. Hence, once the geometry and appearance are learned, one can transform to another articulation state given the part locations and target articulation parameters.

Building on this idea, we frame the learning problem as a conditional novel articulation (and view) synthesis task (see Fig. 1(a)). Given a source observation, multiple views of an object in one articulation state, we first learn the object's shape and appearance by using an implicit model [21] and then freeze its weights. Next we pass the target observation, multi-view images of the same object in a different articulation state, to a tight bottleneck that distills part locations and articulations. We constrain our model to assign each 3D coordinate that is occupied by the object to a part and to apply a valid geometric transformation to the 3D coordinates of each part through ray geometry. The predictions of part segmentation and articulation, along with the target camera viewpoint, are passed to the implicit function and its differential renderer to reproduce the target observations (see Fig. 1(b)). Minimizing the photometric error between the rendered and target view provides supervision for learning part segmentation and articulation. However, joint optimization of these intertwined tasks is challenging and very sensitive to their initialization.

38th Conference on Neural Information Processing Systems (NeurIPS 2024).

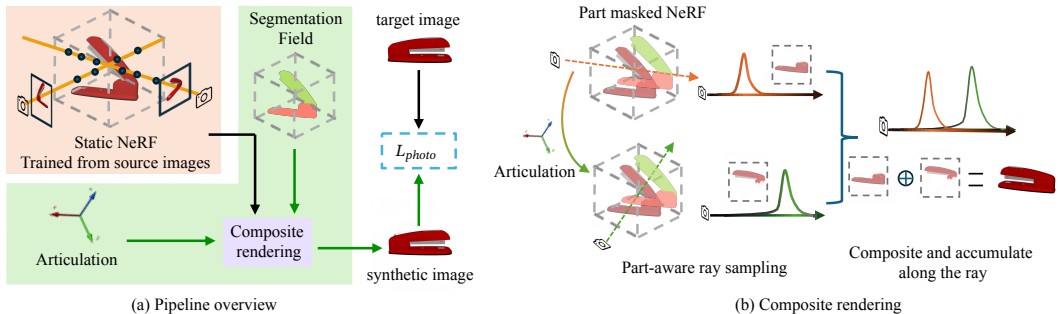

(a) Pipeline overview

(b) Composite rendering

Figure 1: (a) Our method learns the geometry and appearance of an articulated object by first fitting a NeRF from (source) images of an object in a fixed articulation. Then, from another set of (target) images of the object in another articulation, we distill the relative articulation and part labels. Green lines show the gradient path during this distillation. (b) Using the part geometry and appearance from NeRF, we render the target images by compositing the parts after applying the predicted articulations to the segmented parts. The photometric error provides the required supervision for learning the parts and their articulation without groundtruth labels.

To address the optimization challenge, we propose an initialization strategy using an auxiliary voxel grid, which provides an initial estimate for moving parts by computing the errors in the foreground masks when rendering target views for the source articulation. Additionally, we introduce a decoupled optimization procedure that alternates between optimizing the part segmentation on the photometric error and the articulation parameters on the foreground prediction error.

The key advantage of our method, compared to other unsupervised articulation prediction methods [11, 17], is its stable performance across different object and articulation types, its ability to learn from few target views and to model multiple moving parts. Thanks to the stage-wise training, we achieve high-quality object geometry and appearance by using the well-optimized implicit models [2]. Since the part segmentation and articulation parameters form a small portion of the total weights, along with the initialization and decoupled optimization strategies, our method efficiently learns them from few target views, unlike the most relevant work [17] that jointly optimizes multiple part-specific implicit functions from scratch.

## 2 Related work

**Analysis of articulated objects** The analysis of articulated objects typically involves segmentation of movable parts, and estimating their attributes such as position and direction of joint axes, rotation angles, and translation distances. Prior works study articulated objects using 3D input such as meshes [31, 22], point clouds [45, 44, 37, 14] or RGBD images [18, 6, 40, 13] that are error-prone and labor-intensive to collect in real-world scenarios. Recent works [10, 30, 34] that use RGB images to segment 3D planar object parts and estimate their articulation parameters simultaneously require ground-truth labels for segmentation and 3D articulation.

**Articulated object modeling via novel view synthesis** Neural implicit models [20, 26, 21] that are originally designed to model static objects in a 3D consistent way have been extended to articulated objects by multiple recent works [24, 36, 11, 16, 38, 17, 40]. A-SDF [24] learns a disentangled latent space for shape and articulation to synthesize novel shapes and unseen articulated poses for a given object category. CARTO [7] extends A-SDF to multiple object categories and uses a stereo image pairs as input. Similarly, CLA-NeRF [36] learns to perform joint view synthesis, part segmentation, and articulated pose estimation for an object category from multiple views. NARF22 [16] learns a separate neural radiance fields (NeRF) for each part and composes to render the complete object. Unlike our method, both A-SDF and CARTO require 3D shape and articulation pose supervision, CLA-NeRF requires part segmentation labels and is limited to modeling only 1D revolutions for each joint, NARF22 relies on groundtruth part labels and articulation pose.

Most related to our work, DITTO [11], PARIS [17] and Weng et al. [40] aim to estimate part segmentation and articulation pose without labels using from an observation pair of an object in two different articulations. DITTO uses a pair of point cloud as input, can only model shape, whereas both PARIS and our method uses pairs of multi-view images, and to model both geometry and

appearance. PARIS adopts the dynamic/static modelling of [42, 46], learns a separate NeRF for the dynamic and static parts. They are composited using the estimated relative articulation to render the observations with the different articulation. Similarly, [40] first reconstructs the object in two articulation states with two sets of RGBD images. Later, the part-level segmentation is performed based on image registration results. While our method is also based on the same analysis-by-synthesis principle to supervise the training, it differs from previous work in several key ways. Compared to DITTO [11] and Weng et al. [40], our method relies completely on 2D input data. And compared to PARIS [17], our method involves only a single NeRF that is learned on multiple views of an object instance in a fixed articulation pose. Once the NeRF is learned, we freeze its parameters, and learn a segmentation head and articulation to selectively deform the rays while rendering views of different articulations. This means our model's size remain nearly constant when the number of parts increases, which, combined with our two step optimization strategy, leads to more stable and data-efficient learning, yielding significantly better performance. Furthermore, we show that our model goes beyond modeling a single moving object part as in PARIS, and successfully learn multiple moving parts.

**Deformable NeRF** There exists several techniques [27, 29, 35, 28, 5] that model differences between subsequent video frames of a dynamic scene through a deformation field in 3D. While these techniques are general could be used in modeling articulated objects in principle, the deformation field does not provide explicit predictions for part segmentation and articulation. Prior methods that focus on modeling specific articulated and deformable objects such as human body [39, 33, 8, 9, 25] and four-legged animals [41] leverages specialized 3D priors [19] that are not applicable to arbitrary object categories.

## 3 Review: NeRF

Given a set of images $\mathcal{I}$ of an object, a NeRF function [21] maps a tuple $(\boldsymbol{x}, \boldsymbol{d})$, where $\boldsymbol{x} \in \mathbb{R}^3$ is a 3d position and $\boldsymbol{d} \in \mathbb{R}^2$ is a direction, to an RGB color $\boldsymbol{c}$, and a positive scalar volume density $\sigma$. The model outputs volume density $\sigma$ at the location $\boldsymbol{x}$ along with a latent feature vector $\boldsymbol{z}$ which is then concatenated with the viewing direction $\boldsymbol{d}$ and fed into an MLP to estimate the view-dependent RGB color $\boldsymbol{c}$. With a slight abuse of notation, we also use $\sigma$ and $z$ as functions $\sigma(\boldsymbol{x})$ and $z(\boldsymbol{x})$ respectively and color as a function with $c(\boldsymbol{x}, \boldsymbol{d})$. That is, we use boldface to denote vectors.

NeRFs are trained by expressing the color at each pixel of images in the training subset. The color at a pixel $C$ is estimated as $C_N$ via the volume rendering equation as a sum of contributions from points along the ray through the pixel, say $\boldsymbol{r}(t) = o + t\boldsymbol{d}$ where $\boldsymbol{o}$ is the origin of the ray (usually the point of projection of a camera view) and $t$ is a scalar. Points $\boldsymbol{x}_i, i = 1, \ldots, n$ are sampled along the ray as $\boldsymbol{x}_i = \boldsymbol{o} + t_i\boldsymbol{d}$. The estimated color is given by

$$C_N(\boldsymbol{r}) = \sum_{i=1}^{n} T_i^r \left(1 - \exp(-\sigma_i^r \delta_i^r)\mathbf{c}_i^r\right) \quad \text{where} \quad T_i^r = \sum_{j=1}^{i-1} \exp(-\sigma_j^r \delta_j^r) \tag{1}$$

where we use the shorthand notation $\boldsymbol{c}_i^r$ and $\sigma_i^r$ to denote $c(\boldsymbol{x}_i, \boldsymbol{d})$ and $\sigma(\boldsymbol{x}_i)$ respectively, and $\delta_i = t_{i+1} - t_i$ is the distance between samples. And the opacity value is calculated as $O_N(r) = \sum_{i=1}^{n} T_i^r \left(1 - \exp(-\sigma_i^r \delta_i^r)\right)$. The parameters of $c$ and $\sigma$ are obtained by minimizing the photometric loss between predicted color $C_N(\boldsymbol{r})$ and the ground truth color $C(r)$ as

$$\mathcal{L}_{\text{photo}} = \sum_{r \in \mathcal{R}} ||C_N(\boldsymbol{r}) - C(\boldsymbol{r})||_2^2. \tag{2}$$

## 4 Method

Let $\mathcal{I}$ and $\mathcal{I}'$ be two observations of an object with $k$ (known a priori) rigid parts in articulation poses $\mathcal{P}$ and $\mathcal{P}'$ respectively. Each observation contains multiple views of the object along with the foreground masks. We call $\mathcal{I}$ the source observation and $\mathcal{I}'$ the target observation. $\mathcal{P}$ (resp. $\mathcal{P}'$) is a pose tensor composed of $P_\ell \in SE(3)$, $\ell = 1, \ldots, k$ (resp. $P_\ell'$) transformations as $4 \times 4$ matrices corresponding to the local pose of each of the $k$ parts. Our primary goal is to estimate a pose-change tensor $\mathcal{M}$ composed of $M_\ell \in SE(3)$, $\ell = 1, \ldots, k$ so that $P_\ell' = M_\ell P_\ell$. We solve this by starting with the construction of a NeRF from $\mathcal{I}$, as described in Sec. 3 along with the efficient coarse-to-fine

volume sampling strategy in [2], followed by a novel modification to build awareness at the part-level. We use this modified parameterization to optimize an auxiliary voxel grid corresponding to the parts via pose estimation and part segmentation pipelines. Once we have identified the parts and their relative transformations, we are able to render novel views corresponding to articulation pose $\mathcal{P}'$ by transforming view rays suitably by $M_\ell^{-1}$.

## 4.1 Part-aware rendering

Once a NeRF function, which we call 'static NeRF', is learned over $\mathcal{I}$, we freeze its parameters and append a segmentation head $s$ to it towards obtaining part-level information. $s$ is instantiated as 2-layer MLP, and maps the latent feature vector $z(\boldsymbol{x})$ to a probability distribution over the $k$ object parts. We denote the probability of a 3D point $\boldsymbol{x}$ to belong object part $\ell = 1, \ldots, k$ as $s_\ell(z(\boldsymbol{x}))$.

If the segmentation class-probabilities $s_\ell$ and pose-change transformations $M_\ell$ are known, then the object in an unseen articulation pose can be synthesised by suitably transforming the statically trained NeRF without modifying its parameters and then compositing the individual part contributions. To model pose change in each part, we use a different ray, *virtual ray* associated with each part

$$\boldsymbol{r}_\ell = M_\ell^{-1} \boldsymbol{r} \tag{3}$$

and the final rendered color is:

$$C_P(\boldsymbol{r}) = \sum_{i=1}^n \hat{T}_i^r \sum_{\ell=1}^k \left(1 - \exp\left(-\left(s_\ell^{r_\ell}(\boldsymbol{x}_i)\, \sigma^{r_\ell}(\boldsymbol{x}_i)\, \delta_j^{r_\ell}\right)\right)\right) \mathbf{c}_i^{r_\ell} \tag{4}$$

where $\hat{T}_i^r = \sum_{j=1}^{i-1} \exp\left(-\sum_{\ell=1}^k \left(s_\ell^{r_\ell}(\boldsymbol{x}_j)\, \sigma^{r_\ell}(\boldsymbol{x}_j)\, \delta_j^{r_\ell}\right)\right)$ is the transmittance sum. Compared to the original formulation in Eq. (1), it can seen as the density replaced by a part-aware density which is the product of $s$ and $\sigma$. Note that we made a similar modification to the efficient volume sampling strategy of [2] and refer to the Appendix A.1 for more details.

Since the groundtruth part segmentation and articulation are unknown, one could optimize the parameters of $s$ and $\mathcal{M}$ on the target observation such that the photometric error in between the part-aware rendered views and $\mathcal{I}'$. Note that $C_P(\boldsymbol{r})$ is function of $\mathcal{M}$ (through $r_\ell$ in Eq. (3)) and $s$, they can be jointly optimized through backpropagation. However, jointly solving these two intertwined problems through rendering is challenging and highly sensitive to their initial values, as their solutions depend on each other. Hence, we introduce an alternating optimization strategy next.

## 4.2 Decoupled optimization of $s$ and $\mathcal{M}$

To learn the part segmentation and pose-change, we first introduce an auxiliary voxel grid that assigns each 3D coordinate either to a background or a part. Once the voxel is initialized, our training iterates multiple times over a three-step procedure that includes optimization of $\mathcal{M}$, $s$, and refinement of the voxel grid entries.

**Initialization of voxel grid**   Here our key idea is to find the pixel differences between the source and target observations by rendering a target view $v'$ by using the static NeRF, and label the 3D locations of the different pixels as the movable parts to provide a good initialization for estimating the articulation (see Fig. 2). To this end, we first build a 3D voxel of the static NeRF by discretizing its 3D space into 128 bins along each dimension. To compute the voxel entries, we query the density value from the static NeRF at each voxel coordinate and binarize it into occupancy values with criteria $\exp(-\sigma(\boldsymbol{x})\delta) > 0.1$. Next, we use the static NeRF to render opacity and depth images for the viewpoints $v'$. The pixels rendered as foreground in rendered opacity but not in the corresponding foreground mask of $\mathcal{I}'$ will be collected. For those pixels, we further compute their depth by accumulating their density values along the corresponding ray, and use their estimated depth values to tag the occupied voxel entries either with static or dynamic parts. In the case of multiple moving parts, which is assumed to be known, we perform a clustering step [4] to identify $k - 1$ clusters corresponding to moving parts, as we assume that 1 part of the object is static without loss of generality. Finally, we gather the voxel coordinates that is assigned to part $\ell$ to form a matrix of 3D coordinates $X_\ell$.

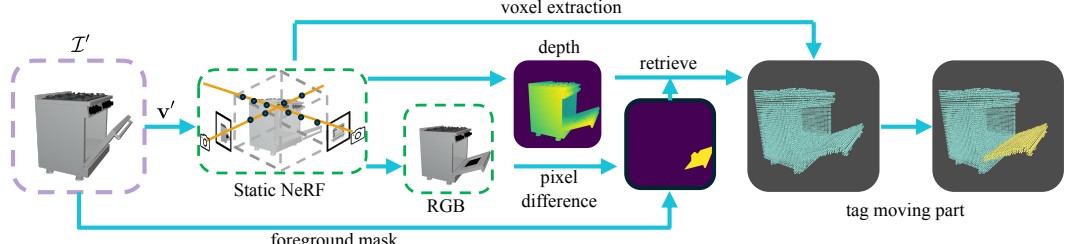

Figure 2: Voxel initialization: identify the voxels belonging to moved parts based on pixel opacity difference.

**Step 1: Optimization of $M_\ell$** As depicted in the right side of Fig. 3, we project the 3D coordinates $X_\ell$ for part $\ell$ onto the image plane for each camera viewpoint $v'$ included in $\mathcal{I}'$ by using $3 \times 4$ dimensional intrinsic camera matrix $K$ as $U_\ell = K M_\ell^{-1} v' X_\ell$, and denote the projected 2D point matrix on the image plane as $U_\ell$. On the left side of Fig. 3, we collect 2D coordinates from the overlap region between rendered opacity and target image $\mathcal{I}'$ to obtain $U'$. Then we concatenate $U'$ and part-specific matrices $U_\ell$ to obtain $U$. We want to optimize $U$ to be the same

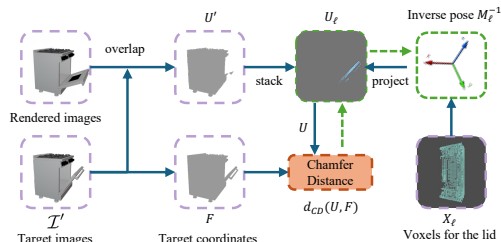

Figure 3: Illustration for optimization of $M_\ell$. The green dotted line shows the gradient flow.

as the target coordinates $F$, which is obtained by gathering the 2D coordinates from the foreground mask of $\mathcal{I}'$. Note that both $U$ and $F$ are matrices where each row corresponds to a 2D image coordinate where $F$ has significantly more rows, as $U$ is obtained from a coarse voxel grid. Then we minimize the 2D Chamfer distance between $U$ and $F$:

$$\mathcal{M}^* = \arg\min_{\mathcal{M}} \ d_{\mathrm{CD}}(U, F). \tag{5}$$

In practice, we only project $X_\ell$ for the moved parts, and stack them with 2D image coordinates of the foreground pixels in both the rendered image and the groundtruth image, as illustrated in Fig. 3.

**Step 2: Optimization of $s_\ell$** Once we obtain the solution of the pose-change $\mathcal{M}^*$ from Step 1, we plug in it to the ray deformation in Eq. (3) and render each view in $\mathcal{I}'$ in Eq. (4). Then we compute photometric loss between the rendered views and $\mathcal{I}'$, and minimize it with respect to the parameters of $s$ only. In the case of multiple moving parts, we initialize the parameters in $s$ using $X_\ell$ for supervision. Please refer to Appendix A.4 for more details.

**Step 3: Voxel grid refinement** The initial part segmentation estimates in the voxel grid are often sparse and noisy due to inaccurate density estimation in NeRF and misassignment of pixels around the foreground and part boundaries. To obtain a more accurate voxel grid, we follow the same steps in the initialization stage except that the 3D coordinates are assigned to the parts based on the predicted label from segmentation head $s$. We denote the new 3D coordinates as $X_\ell^*$. For the consistency, we accept the voxels to $X_\ell^*$ only when the predicted part labels agree within a 3D local neighborhood of $X_\ell$. The neighborhood is defined as being within the distance of one grid cell at a resolution of 128. Note that we only perform the refinement step, after approximately 2k iterations to ensure that the segmentation head produces confident results. Then we also increase the voxel resolution to 256 for each dimension.

## 5 Experiment

### 5.1 Dataset

We evaluate our method on the synthetic 3D PartNet-Mobility dataset [43, 23, 3]. While the dataset contains more than 2000 articulated objects from 46 different categories, we use a subset of the dataset with 6 shapes that was used in [17]. For a fair comparison, we downloaded the processed dataset from [17] which contains 2 sets of 100 views along with their foreground masks, each with a

| Metric | Method | Revolut | | | | Prismatic | |
|---|---|---|---|---|---|---|---|
| | | laptop | oven | stapler | fridge | blade | storage |
| $e_d \downarrow$ | PARIS | $0.68 \pm 0.40$ | $1.04 \pm 0.68$ | $2.42 \pm 0.91$ | $0.81 \pm 0.60$ | $48.58 \pm 25.43$ | $\mathbf{0.34 \pm 0.09}$ |
| | Ours | $\mathbf{0.33 \pm 0.04}$ | $\mathbf{0.34 \pm 0.03}$ | $\mathbf{0.33 \pm 0.04}$ | $\mathbf{0.54 \pm 0.08}$ | $\mathbf{1.54 \pm 0.07}$ | $1.11 \pm 0.14$ |
| $e_p \downarrow$ $(10^{-2})$ | PARIS | $\mathbf{0.18 \pm 0.15}$ | $\mathbf{0.49 \pm 0.53}$ | $55.54 \pm 39.88$ | $\mathbf{0.33 \pm 0.14}$ | - | - |
| | Ours | $0.48 \pm 0.06$ | $1.29 \pm 0.04$ | $\mathbf{0.17 \pm 0.05}$ | $0.44 \pm 0.03$ | - | - |
| $e_g \downarrow$ | PARIS | $0.60 \pm 0.32$ | $0.68 \pm 0.39$ | $44.62 \pm 6.17$ | $0.87 \pm 0.55$ | - | - |
| | Ours | $\mathbf{0.25 \pm 0.03}$ | $\mathbf{0.35 \pm 0.06}$ | $\mathbf{0.290 \pm 0.03}$ | $\mathbf{0.60 \pm 0.05}$ | - | - |
| $e_t \downarrow$ | PARIS | - | - | - | - | $1.13 \pm 0.52$ | $0.30 \pm 0.01$ |
| | Ours | - | - | - | - | $\mathbf{0.01 \pm 0.01}$ | $\mathbf{0.02 \pm 0.03}$ |

Table 1: **Part-level pose estimation results.** Our method outperforms PARIS in majority of object categories while having lower variation over multiple runs in the performance.

different articulation, and also groundtruth part segmentation labels, for each shape. In addition, we select 4 additional shapes, each with two moving parts, and apply the same data generation process to them. Additionally we collected images of a toy car with a handheld device, with camera viewpoint estimated from kiri engine application[15].

We train the static NeRF on 100 views from the first observation, train the part segmentation and articulation on 100 views from the second observation. We report the performance of our method for varying number of views from the second observation in Tab. 4.

Following [17], we report performance in different metrics for pose estimation, novel-view/articulation synthesis, and part segmentation. **Pose estimation:** To report articulation pose performance, we report results in i) direction error $e_d$ that measures the angular discrepancy in degrees between the predicted and actual axis directions across all object categories, ii) position error $e_p$ and geodesic distance $e_d$ for only revolute objects to evaluate the error between the predicted and true axis positions and the error in the predicted rotation angles of parts in degrees, respectively, iii) translation error $e_t$ to measure the disparity between the predicted and actual translation movements for prismatic objects. **Novel-view and -articulation synthesis:** We evaluate the quality of novel view synthesis generated by the models using the Peak Signal-to-Noise Ratio (PSNR) where higher values indicate better reconstruction fidelity. **Part segmentation:** We use mean Intersection over Union (mIoU) on the rendered semantic images to evaluate the accuracy of part segmentation, which is tightly related to the rendered image quality of objects in different articulation states. The ground truth segmentation is generated within the Sapien framework in [43]. Finally, due to the lack of groundtruth segmentation label and articulation in the real data, we only report PSNR and provide qualitative evaluation.

## 5.2 Results

**Baseline** We compare our approach with the state-of-the-art unsupervised technique, PARIS [17] which constructs separate NeRF models for each part of an articulated object and optimizes motion parameters in an end-to-end manner. However, it is limited to two-part objects with only one movable part. As the authors of PARIS do not report their results over multiple runs in their paper, we use their official public code with the default hyperparameters, train both their and our model over 5 random initializations and report the average performance and the standard deviation. We use 2 sets of 100 views for each object to train the models. More implementation details can be found in the supplementary material. We would like to note that the performance of PARIS in our experiments significantly differ from the original results despite all the care taken to reproduce them in the original way[1].

**Part-level pose estimation** For part-level pose estimation, we provide quantitative results in average performance over 5 runs and their standard deviations in Tab. 1, and a qualitative analysis in Fig. 6. The results indicate that our method consistently achieves lower errors across most evaluation metrics

---

[1]Similar problems have been pointed out and the reproducibility of these results has been acknowledged as challenging by the authors in issue 1 and issue 2

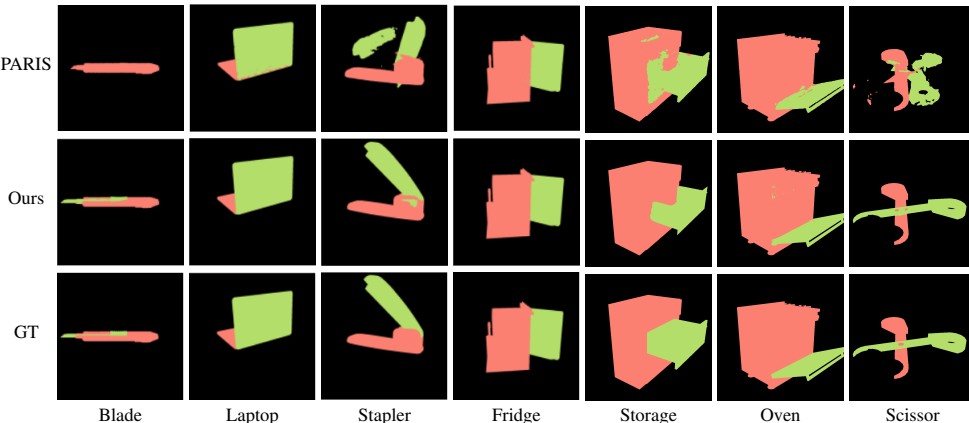

Figure 4: **Qualitative 2D part segmentation results.** Pixels in green denotes the movable parts. Our method demonstrates consistent performance across all tested objects while PARIS failed for Blade, Laptop and Scissor.

compared to PARIS, except the joint position error $e_p$ where the differences are negligible, within the $10^{-3}$ range. Notably, the performance of PARIS on the stapler and blade exhibits significantly higher errors. We observe that PARIS fails to converge to a good solution in all 5 runs for these objects. As shown in Fig. 6, PARIS fails to accurately segment the parts in the stapler and blade, which can also be easily identified in the novel articulation synthesis experiments in Fig. 6. Poor part reconstruction in PARIS results in inaccurate part-level pose estimations. Additionally, the lower standard deviation across all reconstructed objects for our method indicates its stable learning and ability to work for different object instances. We attribute the stability of our method to the decoupled optimization along with the stage-wise training, which contrasts with the joint optimization approach used by PARIS.

**Segmentation and composite rendering** Part segmentation and novel view synthesis in pose $\mathcal{P}'$ are reported in Tab. 2, complemented by a qualitative analysis of part segmentation in Fig. 4. Our method outperforms PARIS across most evaluated objects in part segmentation and image synthesis quality, with the only exception being a minor difference in the PSNR for the laptop.

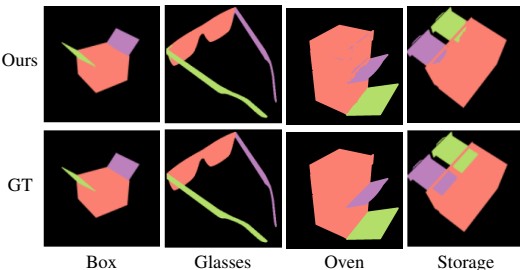

Figure 5: **Qualitative results for 2D multi-part segmentation.** The pink color denotes the static part, while other colors denote the moving parts.

As our model builds on a static NeRF that achieves high quality novel view synthesis for one observation, the rendering quality is largely preserved after the second stage. We provide more detailed analysis in Appendix A.3. Additionally, benefiting from accurate pose estimation via a decoupled approach, our method achieves more robust and precise part segmentation, as depicted in Fig. 4. Here, our method consistently delivers accurate segmentation results for challenging objects such as the blade, stapler, and scissors, where PARIS struggles with accurate part reconstruction. In the other instances including the laptop, storage, and oven, our method achieves visibly better results.

**Evaluation on objects with multiple movable parts** A key advantage of our model is its ability to model objects with multiple moving parts. For such objects, we report pose estimation results in Tab. 3, qualitative for part segmentation in Fig. 5 and novel articulation synthesis in Fig. 6. As in the single part moving object experiments, our method delivers similar performance for the multi-part objects. Notably, we observed a marginally higher joint direction error for glasses, which we attribute to the thin structures such as temples and failure to segment them accurately which can be possibly improved by using higher resolution images.

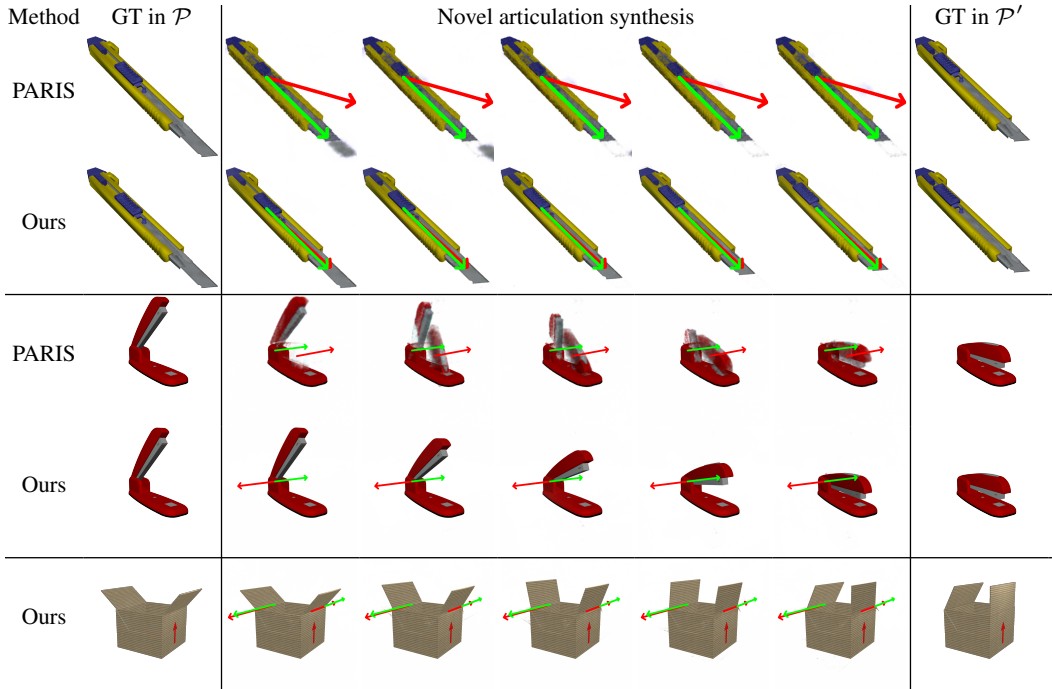

Figure 6: **Qualitative evaluation for novel articulation synthesis.** The ground truth axis is denoted in green and the predicted axis is denoted in red. Please refer to the supplementary for more visualizations.

## 5.3 Ablation studies

We assessed the effectiveness of the proposed decoupled pose estimation of $M_\ell$ (DP) and iterative refinement of $X_\ell$ (IR) on the 'fridge' object. When DP is disabled, the segmentation and articulation are simultaneously learned. In contrast, disabling IR maintains the initial $X_\ell$ for pose estimation. The results in Tab. 5, particularly the first row, demonstrate that joint optimization without DP inaccurately predicts the articulated pose, treating the entire object as static. While enabling DP improves the performance, as shown in the second row, the performance is still poor due to the noisy initial values compared to our full model.

Additionally, we evaluated the impact of the number of views in the target observation for training. We train our model on the randomly subsampled views for multiple runs and averaged their performance (see Tab. 4). Results show a significant drop in pose estimation and rendering quality with 4 images or less. Notably, even with only 8 images, our approach surpasses the performance of PARIS trained with 100 images (indicated with subscript $P$ in Tab. 4). This result clearly show that our method are more robust against fewer viewpoints from the target viewpoints and allows efficiently learning 'articulate' a pretrained NeRF from few target views only.

| Metric | Method | Revolut | | | | Prismatic | |
|---|---|---|---|---|---|---|---|
| | | laptop | oven | stapler | fridge | blade | storage |
| mIoU↑ | PARIS | 0.98 | 0.99 | 0.16 | 0.98 | 0.76 | 0.94 |
| | Ours | **0.99** | **0.99** | **0.98** | **0.99** | **0.94** | **0.96** |
| PSNR ↑ | PARIS | **30.31** | 31.48 | 24.36 | 32.74 | 31.87 | 30.63 |
| | Ours | 29.27 | **32.08** | **34.31** | **35.10** | **36.47** | **34.51** |

Table 2: **Articulation synthesis and part segmentation results.** Average performance over 5 runs ( best results in **boldface**).

| Metric | Revolut | | | Prismatic |
|---|---|---|---|---|
| | oven | glasses | box | storage |
| $e_d \downarrow$ | 1.02 | 2.35 | 0.56 | 1.82 |
| $e_p \downarrow$ | 0.16 | 0.47 | 0.27 | - |
| $e_g \downarrow$ | 1.03 | 1.01 | 0.65 | - |
| $e_t \downarrow (10)$ | - | - | - | 0.11 |
| PSNR ↑ | 32.98 | 29.22 | 28.61 | 28.25 |
| mIoU ↑ | 0.97 | 0.98 | 0.99 | 0.94 |

Table 3: **Objects with multiple parts.** Errors using multiple metrics for pose estimation (averaged over all joints).

| Metric | Num. of images | | | | | | |
|---|---|---|---|---|---|---|---|
| | 2 | 4 | 8 | 16 | 32 | 100 | $100_P$ |
| $e_d \downarrow$ | 46.05 | 8.89 | 0.59 | 0.58 | 0.50 | 0.54 | 0.81 |
| $e_g \downarrow$ | 44.74 | 20.79 | 0.70 | 0.60 | 0.56 | 0.49 | 0.87 |
| PSNR $\uparrow$ | 22.65 | 29.95 | 34.00 | 34.28 | 34.88 | 35.10 | 32.74 |

Table 4: **Ablation studies with different number of target images.**

| Init. | | Metric | | |
|---|---|---|---|---|
| DP | IR | $e_d$ | $e_g$ | PSNR |
| - | - | 2.57 | 26.95 | 18.09 |
| ✓ | - | 2.38 | 14.96 | 25.16 |
| ✓ | ✓ | 0.54 | 0.49 | 35.10 |

Table 5: **Ablation study over different initialization strategies.**

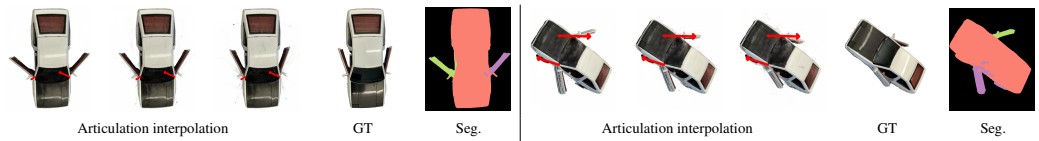

Figure 7: Results on real world examples, the red line indicates the estimated joint axis direction. Green and purple color denotes the moving car door, while pink denotes the body of the toy car. Please refer to Fig. 12 for more qualitative evaluation.

## 5.4 Real-world example

The qualitative evaluation can be found in Fig. 7. Novel articulation synthesis in different views are showed in two sides of Fig. 7. The GT is shown in pose $\mathcal{P}'$ while the segmentation image is the toy car in pose $\mathcal{P}$. The average PSNR we obtain from the static NeRF is , and the PSNR for 28.19, while the PSNR for pose $\mathcal{P}'$ is 24.32. More detail results for segmentaion and articulation interpolation are also presented in Fig. 7.

## 5.5 Limitations

Our method has few limitations too. As we use rendering to supervise our segmentation and pose, our method may fail to segment very thin parts or small movable parts when the rendering error is small. Our method is less stable for multipart objects compared to two-part objects. When the object parts are nearly symmetry, our method may fail to find the correct articulation and choose another articulation that produces similar rendering. We provide examples of such failure cases in Appendix A.3. Additionally, it also inherits the limitations of the implicit models that are built on such as failure to model transparent objects, and inaccurate 3D modeling when the viewpoint information is noisy. Finally, our method is limited to articulated objects with rigid parts and cannot be used to model soft tissue deformations.

## 6 Conclusion

In this study, we tackled the challenges of self-supervised part segmentation, articulated pose estimation, and novel articulation rendering for objects with multiple movable parts. Our approach is the first known attempt to model multipart articulated objects using only multi-view images acquired from two arbitrary articulation states of the same object. We evaluated our method with both synthetic and real-world datasets. The results suggest that our method competently estimates part segmentation and articulation pose parameters, and effectively renders images of unseen articulations, showing promising improvements over existing state-of-the-art techniques. Further, our real-world data experiments underscore the method's robust generalization capabilities. At last, the code and the data used in this project will be released upon acceptance. However, the reliance on geometric information from moving parts for articulation pose estimation poses challenges in modeling highly symmetrical objects. Future work could improve our method by incorporating both appearance and geometric data into the pose estimation process, potentially enhancing accuracy and applicability.

**Broader Impacts** The proposed technique could be potentially used to understand articulated objects for their robotic manipulation in future. The authors are not aware of any potential harm that may arise when the technology is used

**Acknowledgements** Kartic Subr was supported by a Royal Society University Research Fellowship. Hakan Bilen was supported by the EPSRC Visual AI grant EP/T028572/1.

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

| Metric | Method | Revolut | | | | Prismatic | |
|---|---|---|---|---|---|---|---|
| | | laptop | oven | stapler | fridge | blade | storage |
| $e_d \downarrow$ | PARIS[†] | **0.03** | **0.03** | **0.07** | **0.00** | **0.00** | 0.37 |
| | PARIS | 0.18 | 0.48 | 1.30 | 0.22 | 0.27 | **0.24** |
| | Ours | 0.26 | 0.28 | 0.27 | 0.42 | 1.41 | 0.86 |
| $e_p \downarrow$ $(10^{-2})$ | PARIS[†] | **0.01** | **0.03** | **0.06** | **0.02** | - | - |
| | PARIS | 0.08 | 0.04 | 2.12 | 0.17 | - | - |
| | Ours | 0.39 | 1.24 | 0.13 | 0.41 | - | - |
| $e_g \downarrow$ | PARIS[†] | **0.03** | **0.00** | **0.00** | **0.00** | - | - |
| | PARIS | 0.24 | 0.25 | 37.86 | 0.37 | - | - |
| | Ours | 0.22 | 0.31 | 0.24 | 0.55 | - | - |
| $e_t \downarrow (10^{-1})$ | PARIS[†] | - | - | - | - | **0.06** | **0.00** |
| | PARIS | - | - | - | - | 5.84 | 2.93 |
| | Ours | - | - | - | - | 0.12 | 0.05 |

Table 6: Comparison with PARIS in 5-time best setting. PARIS[†] denotes performance reported in the original paper. Best results are shown in **bold**, second best are shown with underline.

# A Appendix / supplemental material

Optionally include supplemental material (complete proofs, additional experiments and plots) in appendix. All such materials **SHOULD be included in the main submission.**

## A.1 Part-aware proposal network

To achieve part-aware composite rendering for articulated objects, we also need to modify the proposal network for correct sampling along the ray $r \in \mathcal{R}'$. In the composite rendering, the proposal network is now required to produce the similar distribution of the weights for samples along the ray. Thus, following the similar design in the part-aware NeRF, we append an extra segmentation field to the proposal network. Now the valid density for each samples becomes segmentation-mask density as $s_\ell^{r_\ell}(\boldsymbol{x}_j)\sigma^{r_\ell}$. The transmittance are then calculated using Eq. (4) without the color term, which will be later used for online distillation. Please refer to the original paper [2] for more details about the online distillation. During the training of part-aware NeRF, the original parameters in the proposal network will be frozen and only updates the segmentation field.

## A.2 Comparison with PARIS

Here we also provide the 5-time best comparison between ours and PARIS. Besides, the reported performance for PARIS is also provided in this subsection. From Tab. 6, we can see that the original reported performance from PARIS are exceptionally better compared to the reproduced ones, even though with the same data and the same configurations. If we only focus on the 5-time best comparison between reproduced PARIS and ours, we can see that we achieve comparable performance. Another thing we can notice is that for prismatic objects, while PARIS has much better estimations on joint axis direction. However, our method later show much better estimation for estimating the moved distance for the dynamic parts. While one thing to notice is that PARIS failed to reconstruct the stapler for all 5 runs in our experiment.

## A.3 Limitations

Details about the limitations and failure cases of our methods.

In our decoupled optimization strategy, the articulated pose estimation relies exclusively on the geometric information from the moving parts. As a result, our method faces challenges with highly

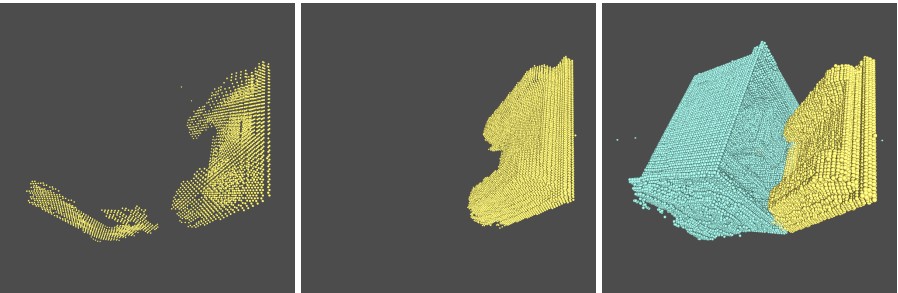

(a) Visualization of dynamic voxel update. From left to right: initialized $X_\ell$, updated $X_\ell$, final $X_\ell$

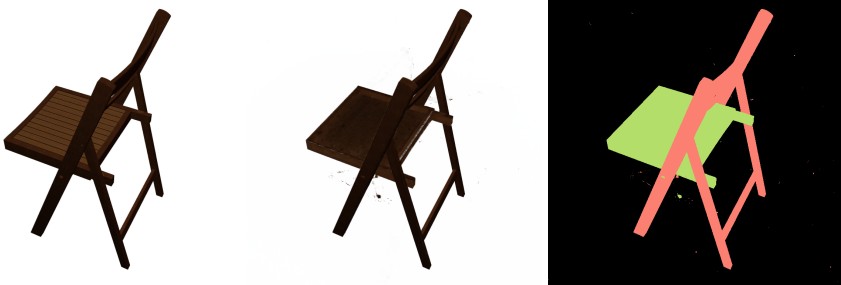

(b) Failure cases for foldchair, from left to right: groundtruth RGB, rendered RGB, part segmentation.

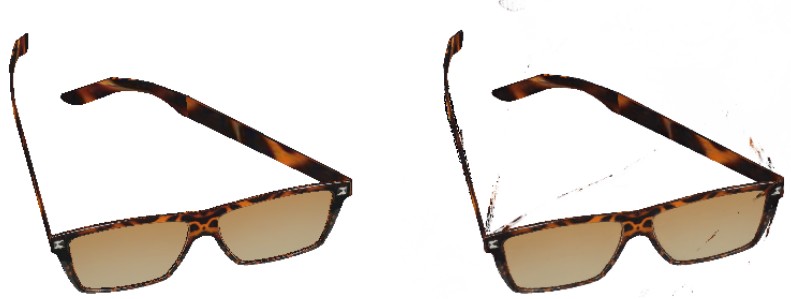

(c) Artifacts for thin parts, the left one is the groundtruth, the right rendering result.

symmetrical components. For example, in the case of a folding chair Appendix A.3, although the segmentation accurately identifies the chair's seat, the pose estimation mistakenly flips the seat by 180 degrees, resulting in the seat being oriented with the bottom upwards as shown in Fig. 8(b). Additionally, our method may encounter difficulties when the articulation motion is minimal, which can lead to insufficient pixel differentiation in images across different articulation poses for initial model estimation. In such cases, both the pose estimator and the iterative updates of $X_\ell$ may underperform or even fail. We also notice that the proposal network have difficulties handling very thin parts in the novel articulation synthesis. As we can see in Appendix A.3, some artifacts appeared at the edge of the frame and the temples for the glasses as in Fig. 8(c), which should be ideally masked out without any opacity and color for those pixels.

Furthermore, the reliance on a pre-trained static NeRF limits the upper boundary of our method's performance in rendering novel articulations. Addressing these limitations could significantly enhance the robustness and applicability of our strategy in handling complex articulated objects. For further analysis of this, please refer to .

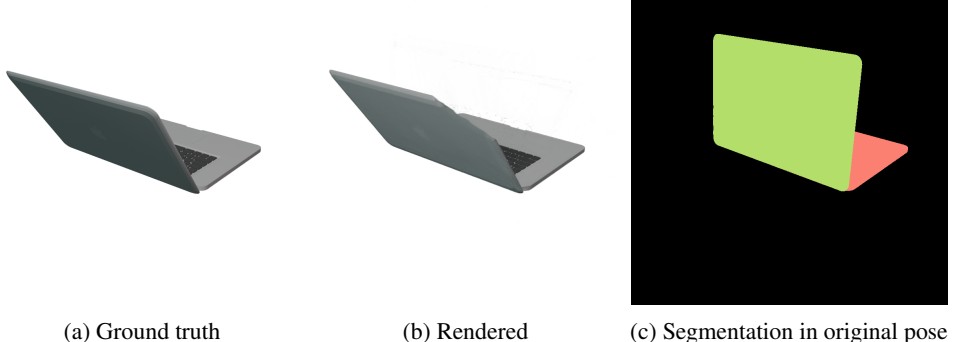

| (a) Ground truth | (b) Rendered | (c) Segmentation in original pose |

Figure 9: We can see in the Fig. 9(b) that the corner of the laptop screen is missing in the novel articulation rendering. While it looks perfect when we check the segmentation in the original pose. Thus, we suspect it is the proposal network than failed to estimate the density distribution for the screen from certain viewpoints.

## A.4 Training setting

As detailed in Sec. 4.2, we perform optimization on $M_\ell$ for 4000 iterations and on $s_\ell$ for 2000 iterations for all evaluated objects. Checkpoints are saved following Step 3. We conduct this process through 5 cycles for objects with a single moving part and 6 cycles for objects with multiple moving parts. Checkpoints with best PSNR during validation will be used for test.

During the first step of $M_\ell$ optimization, we begin with a learning rate of 0.01, which linearly decays by 0.5 every 500 iterations. We accumulate gradients from 8 viewpoints to simulate a batch size of 8, initializing $M_\ell$ identically in the first cycle and using the previously estimated $M_\ell$ for subsequent cycles.

In the second step of $s_\ell$ optimization using the Adam optimizer, the initial learning rate is set at 0.01 and linearly decays by a factor of 0.01 every 100 iterations. For multiple moving parts, initialization involves training the segmentation head $s$ using pre-assigned labels on $X_\ell$, and querying predictions for $\boldsymbol{x} \in X_\ell$. Cross-entropy loss optimized over 1000 iterations with a learning rate of $1e^{-3}$ shapes the learnable parameters in $s$.

Our experiments, requiring around 16 GB of VRAM, complete in approximately 30 minutes on a single RTX 4090 GPU for a single object. As for the training of static NeRF, it takes about 10 minutes with less than 10 GB of VRAM. The estimated total GPU time for this project would be about 2 GPU months.

## A.5 Ablations

### A.5.1 Performance cost for novel articulation synthesis

We also provide an ablation study to investigate the performance drops for conditional novel articulation synthesis compared to the pre-trained static model. As shown in Tab. 7, the performance drop is subject to different category of objects ranging from smallest 13.3% for the fridge and blade to 31.4% laptop. We suspect the significant drops for laptop is caused by the deteriorate performance of the proposal network on the thin laptop screen. Visualizations can be found in Fig. 9. The results indicate that our method can benefit from the high quality of appearance reconstruction from the pre-trained static NeRF.

## A.6 Visualization

Here we show more visualization for qualitative evaluations. Besides, additional animated images for the real-world toy car is also provided in the supplementary materials.

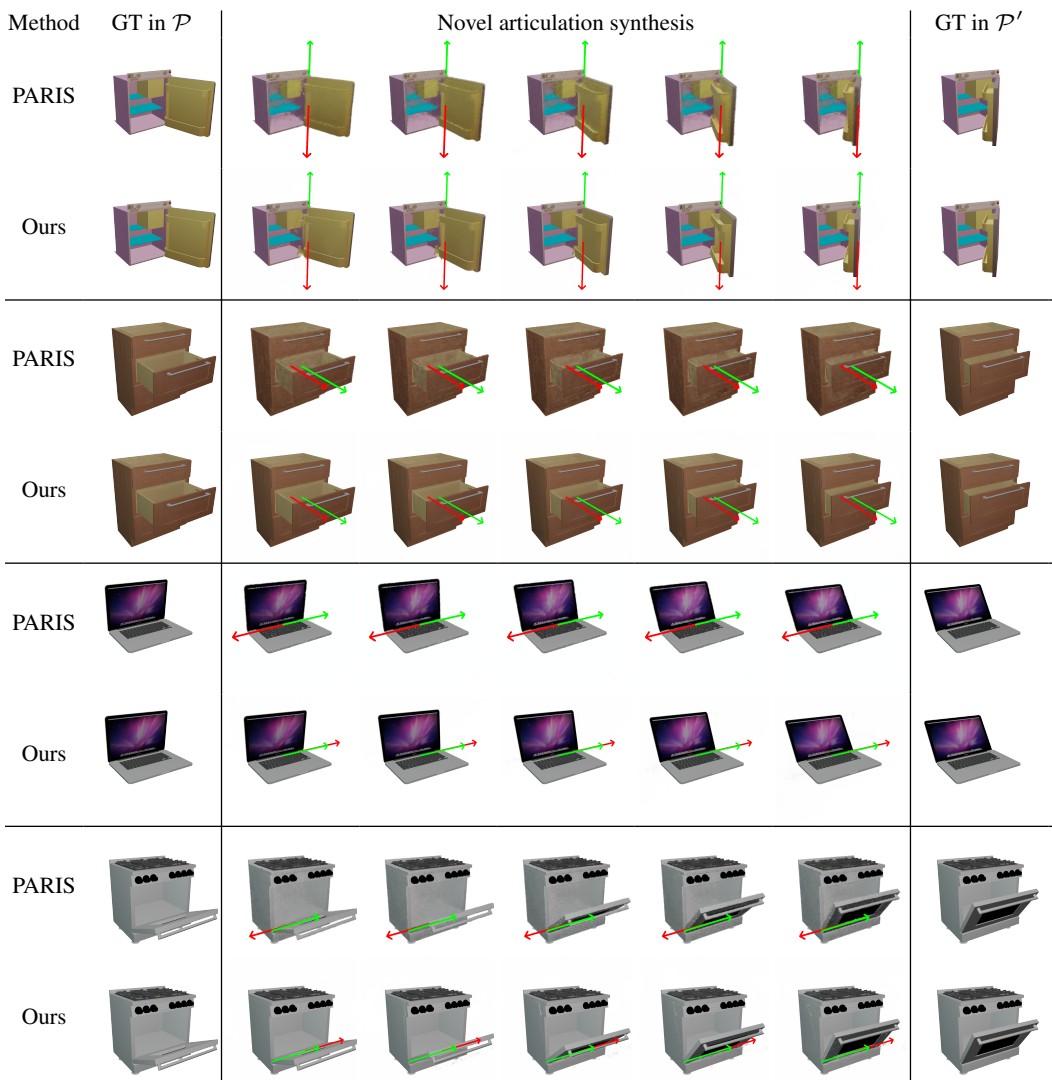

Figure 10: **Articulation interpolation for single moving part objects.**

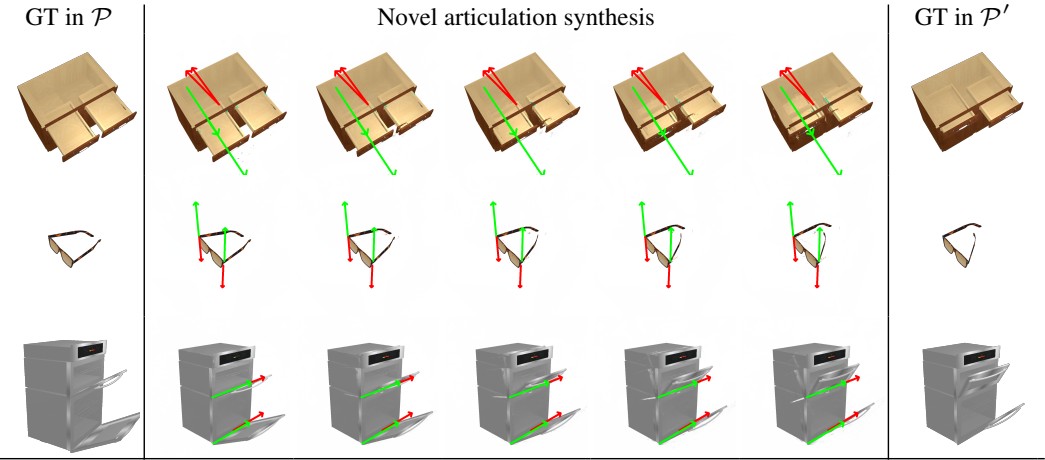

Figure 11: **Articulation interpolation for multiple moving part objects.**

| Method | Object | | | | | | | | | |
|--------|--------|------|---------|--------|-------|---------|-------|---------|-------|----------|
|        | laptop | oven | stapler | fridge | blade | storage | oven* | glasses* | box* | storage* |
| Static | 42.70 | 39.69 | 42.39 | 40.50 | 42.7 | 40.33 | 39.03 | 37.93 | 36.56 | 35.23 |
| Art. | 29.27 | 32.08 | 34.31 | 35.1 | 36.47 | 34.51 | 32.98 | 29.22 | 28.61 | 28.25 |
| Δ | -31.4% | -19.2% | -19.1% | -13.30% | -13.3% | -14.4% | -15.5% | -22.8% | -21.8% | -19.8% |

Table 7: Comparison of rendering quality between objects in their original pose $\mathcal{P}$ and articulated pose $\mathcal{P}'$. 'Static' refers to the rendering performance of the object in its original pose $\mathcal{P}$, whereas 'Art.' indicates the rendering quality of the object in articulated pose $\mathcal{P}'$ using our method with the static NeRF. Objects marked with * represent those with multiple movable parts.

|  | $e_d \downarrow$ | $e_p \downarrow$ | $e_g \downarrow$ | $PSNR \uparrow$ | $mIoU \uparrow$ |
|--|------------------|------------------|------------------|-----------------|-----------------|
| PARIS | $1.96 \pm 1.14$ | $0.27 \pm 0.13$ | $2.04 \pm 0.62$ | $30.97 \pm 0.83$ | $0.95 \pm 0.01$ |
| Ours | $\mathbf{0.25 \pm 0.01}$ | $\mathbf{0.06 \pm 0.01}$ | $\mathbf{0.80 \pm 0.19}$ | $\mathbf{32.27 \pm 0.29}$ | $\mathbf{0.96 \pm 0.02}$ |

Table 8: Quantitative evaluation for the same door instance. We report average performance over 5 runs. For the description of $e_d$, $e_p$, $e_g$, $PSNR$ and $mIoU$, please see Section 5.1 in the submission. Our method outperforms PARIS consistently in all the metrics.

## A.7 More results

Novel articulation synthesis

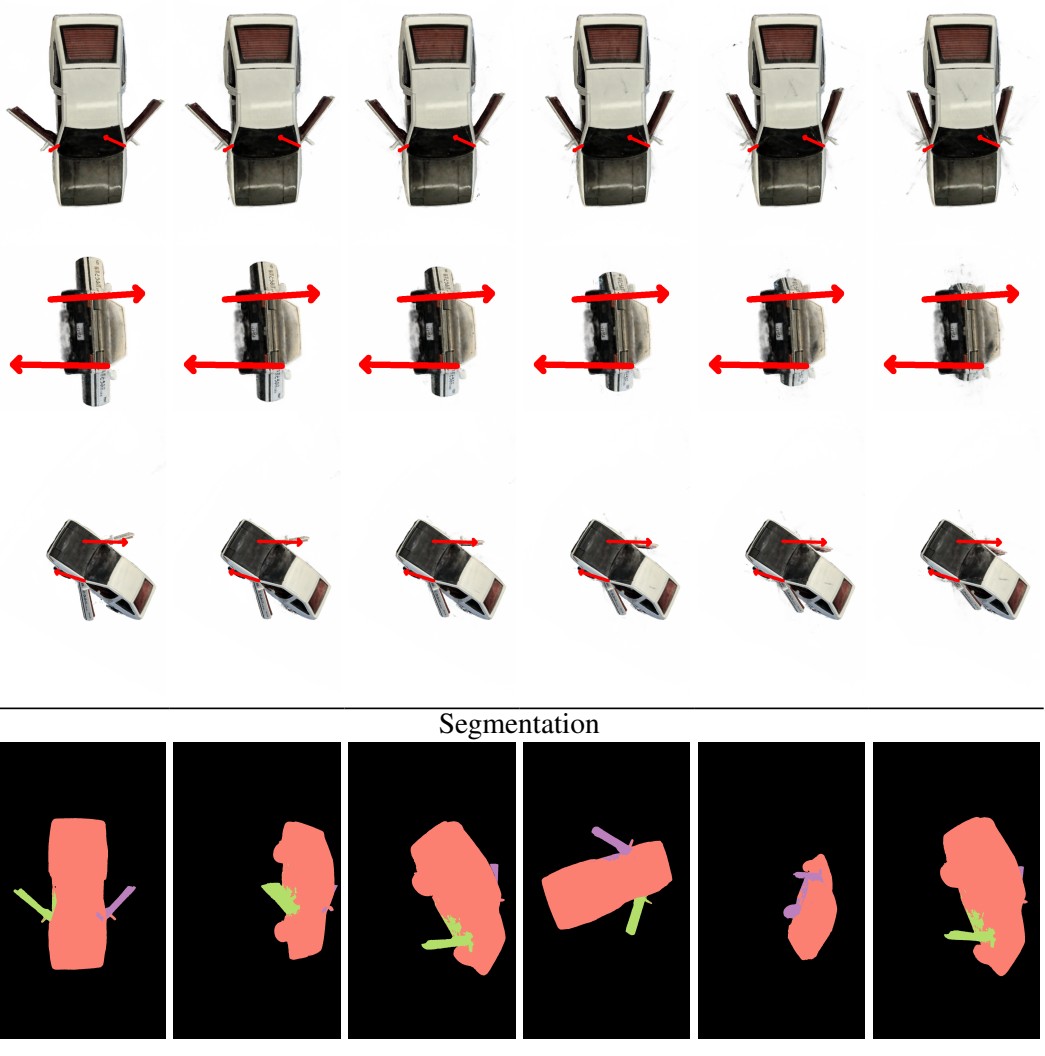

Segmentation

Figure 12: **More Qualitative evaluation on real-world object.**


Figure 13: Here we show qualitative results for a 'door' instance along with its frame. In contrast to the instances in our submission, the static part (frame) is smaller than the moving part (door). Given two sets of views in the articulation $P$ and $P'$, we provide the original input images and their rendering in the respective articulations, ground-truth (GT) and predicted part segmentation results. Our method achieves faithful rendering results in different articulations with minor artifacts and accurate part segmentation.

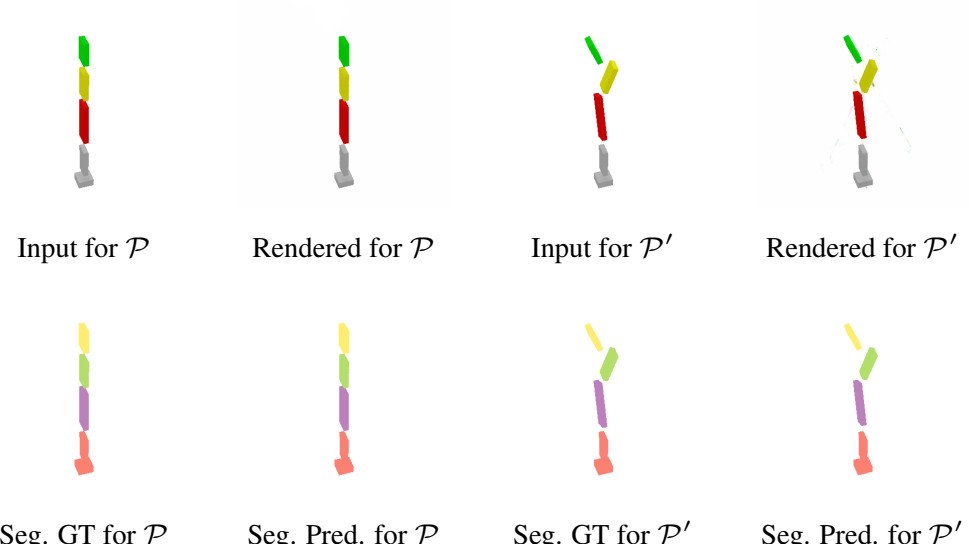

Input for $\mathcal{P}$  Rendered for $\mathcal{P}$  Input for $\mathcal{P}'$  Rendered for $\mathcal{P}'$

Seg. GT for $\mathcal{P}$  Seg. Pred. for $\mathcal{P}$  Seg. GT for $\mathcal{P}'$  Seg. Pred. for $\mathcal{P}'$

Figure 14: Here we show qualitative results for a toy robotic arm with 2 static joints (grey and blue) and 3 moving joints (red, yellow and green). Unlike the objects in the Sapiens dataset that have independently moving parts, the joints of the robotic arm are linearly connected and forming a chain structure. Note that our method independently estimates the 6 DoF $SE(3)$ transformation for each moving part between articulation pose $\mathcal{P}$ and $\mathcal{P}'$. Given the kinematic structure, the local transformations could be computed. Our method faithfully renders the robotic arm in different poses with minor artifacts, and segments its parts accurately. Note that the static parts are labeled as one part.

- The claims made should match theoretical and experimental results, and reflect how much the results can be expected to generalize to other settings.
- It is fine to include aspirational goals as motivation as long as it is clear that these goals are not attained by the paper.

2. **Limitations**

Question: Does the paper discuss the limitations of the work performed by the authors?

Answer: [Yes]

Justification: Details about discussions about limitations can be found in Sec. 5.

Guidelines:

- The answer NA means that the paper has no limitation while the answer No means that the paper has limitations, but those are not discussed in the paper.
- The authors are encouraged to create a separate "Limitations" section in their paper.
- The paper should point out any strong assumptions and how robust the results are to violations of these assumptions (e.g., independence assumptions, noiseless settings, model well-specification, asymptotic approximations only holding locally). The authors should reflect on how these assumptions might be violated in practice and what the implications would be.
- The authors should reflect on the scope of the claims made, e.g., if the approach was only tested on a few datasets or with a few runs. In general, empirical results often depend on implicit assumptions, which should be articulated.
- The authors should reflect on the factors that influence the performance of the approach. For example, a facial recognition algorithm may perform poorly when image resolution is low or images are taken in low lighting. Or a speech-to-text system might not be used reliably to provide closed captions for online lectures because it fails to handle technical jargon.
- The authors should discuss the computational efficiency of the proposed algorithms and how they scale with dataset size.
- If applicable, the authors should discuss possible limitations of their approach to address problems of privacy and fairness.
- While the authors might fear that complete honesty about limitations might be used by reviewers as grounds for rejection, a worse outcome might be that reviewers discover limitations that aren't acknowledged in the paper. The authors should use their best judgment and recognize that individual actions in favor of transparency play an important role in developing norms that preserve the integrity of the community. Reviewers will be specifically instructed to not penalize honesty concerning limitations.

3. **Theory Assumptions and Proofs**

Question: For each theoretical result, does the paper provide the full set of assumptions and a complete (and correct) proof?

Answer: [NA]

Justification: In this paper, we focus more on the practical results, which is shown in Sec. 5.

Guidelines:

- The answer NA means that the paper does not include theoretical results.
- All the theorems, formulas, and proofs in the paper should be numbered and cross-referenced.
- All assumptions should be clearly stated or referenced in the statement of any theorems.
- The proofs can either appear in the main paper or the supplemental material, but if they appear in the supplemental material, the authors are encouraged to provide a short proof sketch to provide intuition.
- Inversely, any informal proof provided in the core of the paper should be complemented by formal proofs provided in appendix or supplemental material.
- Theorems and Lemmas that the proof relies upon should be properly referenced.

4. **Experimental Result Reproducibility**

Question: Does the paper fully disclose all the information needed to reproduce the main experimental results of the paper to the extent that it affects the main claims and/or conclusions of the paper (regardless of whether the code and data are provided or not)?

Answer: [Yes]

Justification: Yes, the details about our method can be found in Sec. 4.

Guidelines:

- The answer NA means that the paper does not include experiments.
- If the paper includes experiments, a No answer to this question will not be perceived well by the reviewers: Making the paper reproducible is important, regardless of whether the code and data are provided or not.
- If the contribution is a dataset and/or model, the authors should describe the steps taken to make their results reproducible or verifiable.
- Depending on the contribution, reproducibility can be accomplished in various ways. For example, if the contribution is a novel architecture, describing the architecture fully might suffice, or if the contribution is a specific model and empirical evaluation, it may be necessary to either make it possible for others to replicate the model with the same dataset, or provide access to the model. In general. releasing code and data is often one good way to accomplish this, but reproducibility can also be provided via detailed instructions for how to replicate the results, access to a hosted model (e.g., in the case of a large language model), releasing of a model checkpoint, or other means that are appropriate to the research performed.
- While NeurIPS does not require releasing code, the conference does require all submissions to provide some reasonable avenue for reproducibility, which may depend on the nature of the contribution. For example
  (a) If the contribution is primarily a new algorithm, the paper should make it clear how to reproduce that algorithm.
  (b) If the contribution is primarily a new model architecture, the paper should describe the architecture clearly and fully.
  (c) If the contribution is a new model (e.g., a large language model), then there should either be a way to access this model for reproducing the results or a way to reproduce the model (e.g., with an open-source dataset or instructions for how to construct the dataset).
  (d) We recognize that reproducibility may be tricky in some cases, in which case authors are welcome to describe the particular way they provide for reproducibility. In the case of closed-source models, it may be that access to the model is limited in some way (e.g., to registered users), but it should be possible for other researchers to have some path to reproducing or verifying the results.

5. **Open access to data and code**

Question: Does the paper provide open access to the data and code, with sufficient instructions to faithfully reproduce the main experimental results, as described in supplemental material?

Answer: [Yes]

Justification: Code and data will be released upon acceptance.

Guidelines:

- The answer NA means that paper does not include experiments requiring code.
- Please see the NeurIPS code and data submission guidelines (`https://nips.cc/public/guides/CodeSubmissionPolicy`) for more details.
- While we encourage the release of code and data, we understand that this might not be possible, so "No" is an acceptable answer. Papers cannot be rejected simply for not including code, unless this is central to the contribution (e.g., for a new open-source benchmark).
- The instructions should contain the exact command and environment needed to run to reproduce the results. See the NeurIPS code and data submission guidelines (`https://nips.cc/public/guides/CodeSubmissionPolicy`) for more details.

- The authors should provide instructions on data access and preparation, including how to access the raw data, preprocessed data, intermediate data, and generated data, etc.
- The authors should provide scripts to reproduce all experimental results for the new proposed method and baselines. If only a subset of experiments are reproducible, they should state which ones are omitted from the script and why.
- At submission time, to preserve anonymity, the authors should release anonymized versions (if applicable).
- Providing as much information as possible in supplemental material (appended to the paper) is recommended, but including URLs to data and code is permitted.

6. **Experimental Setting/Details**

   Question: Does the paper specify all the training and test details (e.g., data splits, hyper-parameters, how they were chosen, type of optimizer, etc.) necessary to understand the results?

   Answer: [Yes]

   Justification: Details on the training setting is shown in Appendix A.4.

   Guidelines:

   - The answer NA means that the paper does not include experiments.
   - The experimental setting should be presented in the core of the paper to a level of detail that is necessary to appreciate the results and make sense of them.
   - The full details can be provided either with the code, in appendix, or as supplemental material.

7. **Experiment Statistical Significance**

   Question: Does the paper report error bars suitably and correctly defined or other appropriate information about the statistical significance of the experiments?

   Answer: [Yes]

   Justification: We ran our experiments multiple times and report average performance with the standard deviation. In addition, we report results over a set of diverse objects and evaluate them by using several metrics.

   Guidelines:

   - The answer NA means that the paper does not include experiments.
   - The authors should answer "Yes" if the results are accompanied by error bars, confidence intervals, or statistical significance tests, at least for the experiments that support the main claims of the paper.
   - The factors of variability that the error bars are capturing should be clearly stated (for example, train/test split, initialization, random drawing of some parameter, or overall run with given experimental conditions).
   - The method for calculating the error bars should be explained (closed form formula, call to a library function, bootstrap, etc.)
   - The assumptions made should be given (e.g., Normally distributed errors).
   - It should be clear whether the error bar is the standard deviation or the standard error of the mean.
   - It is OK to report 1-sigma error bars, but one should state it. The authors should preferably report a 2-sigma error bar than state that they have a 96% CI, if the hypothesis of Normality of errors is not verified.
   - For asymmetric distributions, the authors should be careful not to show in tables or figures symmetric error bars that would yield results that are out of range (e.g. negative error rates).
   - If error bars are reported in tables or plots, The authors should explain in the text how they were calculated and reference the corresponding figures or tables in the text.

8. **Experiments Compute Resources**

   Question: For each experiment, does the paper provide sufficient information on the computer resources (type of compute workers, memory, time of execution) needed to reproduce the experiments?

Answer: [Yes]

Justification: We report the compute resources used in our work in Appendix A.4.

Guidelines:

- The answer NA means that the paper does not include experiments.
- The paper should indicate the type of compute workers CPU or GPU, internal cluster, or cloud provider, including relevant memory and storage.
- The paper should provide the amount of compute required for each of the individual experimental runs as well as estimate the total compute.
- The paper should disclose whether the full research project required more compute than the experiments reported in the paper (e.g., preliminary or failed experiments that didn't make it into the paper).

9. **Code Of Ethics**

Question: Does the research conducted in the paper conform, in every respect, with the NeurIPS Code of Ethics `https://neurips.cc/public/EthicsGuidelines`?

Answer: [Yes]

Justification: This project conform with the NeurIPS Code of Ethics.

Guidelines:

- The answer NA means that the authors have not reviewed the NeurIPS Code of Ethics.
- If the authors answer No, they should explain the special circumstances that require a deviation from the Code of Ethics.
- The authors should make sure to preserve anonymity (e.g., if there is a special consideration due to laws or regulations in their jurisdiction).

10. **Broader Impacts**

Question: Does the paper discuss both potential positive societal impacts and negative societal impacts of the work performed?

Answer: [NA]

Justification: We don't discuss the social impact of this project in the paper.

Guidelines:

- The answer NA means that there is no societal impact of the work performed.
- If the authors answer NA or No, they should explain why their work has no societal impact or why the paper does not address societal impact.
- Examples of negative societal impacts include potential malicious or unintended uses (e.g., disinformation, generating fake profiles, surveillance), fairness considerations (e.g., deployment of technologies that could make decisions that unfairly impact specific groups), privacy considerations, and security considerations.
- The conference expects that many papers will be foundational research and not tied to particular applications, let alone deployments. However, if there is a direct path to any negative applications, the authors should point it out. For example, it is legitimate to point out that an improvement in the quality of generative models could be used to generate deepfakes for disinformation. On the other hand, it is not needed to point out that a generic algorithm for optimizing neural networks could enable people to train models that generate Deepfakes faster.
- The authors should consider possible harms that could arise when the technology is being used as intended and functioning correctly, harms that could arise when the technology is being used as intended but gives incorrect results, and harms following from (intentional or unintentional) misuse of the technology.
- If there are negative societal impacts, the authors could also discuss possible mitigation strategies (e.g., gated release of models, providing defenses in addition to attacks, mechanisms for monitoring misuse, mechanisms to monitor how a system learns from feedback over time, improving the efficiency and accessibility of ML).

11. **Safeguards**

Question: Does the paper describe safeguards that have been put in place for responsible release of data or models that have a high risk for misuse (e.g., pretrained language models, image generators, or scraped datasets)?

Answer: [Yes]

Justification: There are no such risks in this paper.

Guidelines:

- The answer NA means that the paper poses no such risks.
- Released models that have a high risk for misuse or dual-use should be released with necessary safeguards to allow for controlled use of the model, for example by requiring that users adhere to usage guidelines or restrictions to access the model or implementing safety filters.
- Datasets that have been scraped from the Internet could pose safety risks. The authors should describe how they avoided releasing unsafe images.
- We recognize that providing effective safeguards is challenging, and many papers do not require this, but we encourage authors to take this into account and make a best faith effort.

12. **Licenses for existing assets**

Question: Are the creators or original owners of assets (e.g., code, data, models), used in the paper, properly credited and are the license and terms of use explicitly mentioned and properly respected?

Answer: [Yes]

Justification: We properly cited the origianl paper for the SAPIEN synthetic dataset used in our paper. The asset for SAPIEN follows the MIT license.

Guidelines:

- The answer NA means that the paper does not use existing assets.
- The authors should cite the original paper that produced the code package or dataset.
- The authors should state which version of the asset is used and, if possible, include a URL.
- The name of the license (e.g., CC-BY 4.0) should be included for each asset.
- For scraped data from a particular source (e.g., website), the copyright and terms of service of that source should be provided.
- If assets are released, the license, copyright information, and terms of use in the package should be provided. For popular datasets, `paperswithcode.com/datasets` has curated licenses for some datasets. Their licensing guide can help determine the license of a dataset.
- For existing datasets that are re-packaged, both the original license and the license of the derived asset (if it has changed) should be provided.
- If this information is not available online, the authors are encouraged to reach out to the asset's creators.

13. **New Assets**

Question: Are new assets introduced in the paper well documented and is the documentation provided alongside the assets?

Answer: [Yes]

Justification: Codes and data will be released upon acceptance, together with instructions for use and license.

Guidelines:

- The answer NA means that the paper does not release new assets.
- Researchers should communicate the details of the dataset/code/model as part of their submissions via structured templates. This includes details about training, license, limitations, etc.
- The paper should discuss whether and how consent was obtained from people whose asset is used.

- At submission time, remember to anonymize your assets (if applicable). You can either create an anonymized URL or include an anonymized zip file.

14. **Crowdsourcing and Research with Human Subjects**

   Question: For crowdsourcing experiments and research with human subjects, does the paper include the full text of instructions given to participants and screenshots, if applicable, as well as details about compensation (if any)?

   Answer: [NA]

   Justification: This paper focuses on non-living objects.

   Guidelines:

   - The answer NA means that the paper does not involve crowdsourcing nor research with human subjects.
   - Including this information in the supplemental material is fine, but if the main contribution of the paper involves human subjects, then as much detail as possible should be included in the main paper.
   - According to the NeurIPS Code of Ethics, workers involved in data collection, curation, or other labor should be paid at least the minimum wage in the country of the data collector.

15. **Institutional Review Board (IRB) Approvals or Equivalent for Research with Human Subjects**

   Question: Does the paper describe potential risks incurred by study participants, whether such risks were disclosed to the subjects, and whether Institutional Review Board (IRB) approvals (or an equivalent approval/review based on the requirements of your country or institution) were obtained?

   Answer: [NA]

   Justification: This paper focuses on non-living objects.

   Guidelines:

   - The answer NA means that the paper does not involve crowdsourcing nor research with human subjects.
   - Depending on the country in which research is conducted, IRB approval (or equivalent) may be required for any human subjects research. If you obtained IRB approval, you should clearly state this in the paper.
   - We recognize that the procedures for this may vary significantly between institutions and locations, and we expect authors to adhere to the NeurIPS Code of Ethics and the guidelines for their institution.
   - For initial submissions, do not include any information that would break anonymity (if applicable), such as the institution conducting the review.

