# OpenReview forum: "Articulate your NeRF: Unsupervised articulated object modeling via conditional view synthesis"
_NeurIPS.cc/2024/Conference — NeurIPS 2024 poster_

### Official Review · Reviewer_AUv9 · 2024-07-02

**Soundness:** 2
**Presentation:** 1
**Contribution:** 1
**Rating:** 2
**Confidence:** 4

**Summary:**

This paper presents an unsupervised method to learn the pose and part-segmentation of articulated objects with rigid parts using conditional view synthesis. Their approach learns the NeRF of both states and extracts 3D voxels for optimizing the pose and segmentation.

**Strengths:**

Using 3D Voxel instead of mesh provides another approach.

**Weaknesses:**

Firstly, my main concern is about the novelty and contribution of this work.

Compared to the work PARIS, which this paper frequently references, I did not find any significant innovative points. In fact, the method appears to have regressed. This method separates the optimization of pose and segmentation, first optimizing the pose and then optimizing the segmentation. In my opinion, this is not as effective as jointly optimizing them, since the segmentation results can also affect the pose optimization results. However, the 3D voxel approach used in this paper is not differentiable (Lines 150-151), so it cannot jointly optimize both.

And the voxel grid refinement operation seems more like a simple engineering post-processing.

Additionally, I have some issues with the experimental results. The values reported for the existing method PARIS in this paper differ from the results reported in the original paper. Have the authors considered comparing their method with other existing methods?

**Questions:**

Please see the Weakness section.

**Limitations:**

I agree with the limitation section, but, I find the method for finding U (Line 170) to be insufficiently robust. If an issue arises, all subsequent operations will fail.

---

> ### Author Rebuttal · Authors · 2024-08-07
>
> Thanks for reviewing our paper.
>
> 1. **Alternating optimization:** ```This method separates the optimization of pose and segmentation, first optimizing the pose and then optimizing the segmentation. In my opinion, this is not as effective as jointly optimizing them, since the segmentation results can also affect the pose optimization results.```
>
> We agree with the reviewer that the pose and segmentation solutions depend on each other and, however, we disagree that joint optimization is guaranteed to obtain better results. It is well known that simultaneously updating intertwined parameters can be challenging and result in poor solutions. Hence, we use a well-known principled optimization strategy, coordinate block descent [R1] that optimizes first pose and then segmentation iteratively. We also quantitatively supported our argument that our method with the suggested joint optimization (without the decoupled pose estimation (DP) and iterative refinement (IR), the first row in Table 5) performs poorly. Here, the segmentation head and the articulated pose tensor are jointly optimized during the training (Line 276 - 277).
>
> [R1] Wright, Stephen J. "Coordinate descent algorithms." Mathematical programming 151.1 (2015): 3-34.
>
>
>
> 2. **Novelty:** ```Compared to the work PARIS, which this paper frequently references, I did not find any significant innovative points.```
>
> We refer the reviewer to the originality point in the reviewer guidelines of the conference and invite the reviewer to provide a more substantiated comment on their understanding of “significant innovation”. Our method significantly differs from PARIS in three ways. First, it only requires a single neural radiance field (NeRF) for each object, while PARIS requires one for each part. Second, our method can model objects with multiple moving parts by segmenting NeRF space, while PARIS is limited to two part objects (with one as static and the other one as dynamic). Third, our method performs consistently across different objects unlike PARIS thanks to the proposed initialization strategy and iterative 3-step optimization. Clearly, our method is composed of a novel combination of multiple steps which has not been applied to the related problems, and significantly differs from the prior work in terms of methodology and capability.
>
>
> 3. **Non-differentiable representation:** ```the 3D voxel approach used in this paper is not differentiable (Lines 150-151), so it cannot jointly optimize both.```
>
> The 3D voxel strategy is used only for initializing the optimization (see Line 150-151). Therefore, it is not required to be differential. While the initialization can be noisy, we iteratively refine it by acquiring estimates from the segmentation head later and demonstrate its effectiveness in Table 5.
>
>
> 4. **Simple post-processing:** ```And the voxel grid refinement operation seems more like a simple engineering post-processing.```
>
> The voxel grid refinement is not a post-processing step but an update procedure in the optimization that updates the 3D coordinates based on the estimation from the segmentation head to enable more accurate pose estimation in the next iteration. Please refer to line 185-194 for more details.
>
>
>
>
> 5. **Reproducing the prior work:** ```The values reported for the existing method PARIS in this paper differ from the results reported in the original paper.”```
>
> The reviewer missed that we already pointed out the reproducibility issue of PARIS in the footnote of page 6. Similar problems for reproducibility for PARIS have been acknowledged as challenging by other research in issues on the official repo. The author’s response to the reproducibility issue in the official code repo is:
>
> ```“the training for each object is essentially an optimization process, which can be easily affected by the randomness. I was also encountering the unstable training issue you mentioned in my practice”```
>
> For a fair comparison, we follow the default setting provided from the official repo, and reproduce the result on the officially released data. To account for the randomness, we report mean and std metric for 5 independent runs in our paper. In this setting, we can see that our methods are more robust across different objects and delivered more consistent performance compared to PARIS.
>
> Besides, concurrent work [R2] also reports the reproduced performance of PARIS using the officially released data, which also shows similarly lower performance for PARIS.
>
> [R2] Neural Implicit Representation for Building Digital Twins of Unknown Articulated Objects, CVPR 2024
>
>
> 6. **Comparison to other works:** ```Have the authors considered comparing their method with other existing methods?```
>
> We have compared our method to the most recent and related work, PARIS. If the reviewer is aware of other related work which was not discussed in our submission, we are happy to provide further discussion.
>
> 7. **Robustness:** ```I find the method for finding U (Line 170) to be insufficiently robust. If an issue arises, all subsequent operations will fail.```
>
> First, the U mentioned in line 170 refers to the projected 2D points on the image space, which is obtained using standard camera projection formula from 3D points $X_\ell$ but not found by our method like the 3D coordinates $X_\ell$. If the reviewer is referring to the 3D coordinates, we already provided rigorous evaluation of our method on multiple categories of objects and also further study the robustness of our method in case of noisy and incomplete initialization. We refer to Figure 2 and Figure 8 for more details.

---

> > ### Comment · Reviewer_AUv9 · 2024-08-11
> > **Feedback**
> >
> > I still do not believe that this work offers sufficient innovation and contribution compared to existing works, and there are significant issues with the experimental section.
> >
> > Regarding your statement: "we disagree that joint optimization is guaranteed to obtain better results. It is well known that simultaneously updating intertwined parameters can be challenging and result in poor solutions," I expect to see comparative experimental results or valid references to support this claim rather than an assertion without evidence. Furthermore, if both the segmentation process and the transformation process are differentiable, we can still iteratively update both. I do not believe that replacing a differentiable process with a non-differentiable one, relying on iterative approximation is a good idea.
> >
> > The voxel grid refinement operation is not a one-time post possessing, while it is still a simple engineering add-on.
> >
> > As you pointed out in your CVPR2024 review R2, I do not consider this work to be concurrent, as it was already published before your submission. Both R2 and the works they compared can be used as benchmarks for comparison. Since you found that the PARIS work could not be reproduced, it is clearly unacceptable to compare and only compare it against poorly reproduced results.

---

> ### Author Response · Authors · 2024-08-12
>
> 1. **Sufficient innovation and contribution compared to existing works:** We refer the reviewer to the originality point in the reviewer guidelines of the conference and invite the reviewer to provide a more substantiated comment on their understanding of “significant innovation”.
> 2. **Comparative experimental results:** The reviewer missed the point in our response. We would like to reiterate that our method without the decoupled pose estimation (DP) and iterative refinement (IR), which is reported at the first row of Table 5, is the joint optimization baseline that is requested by the reviewer. This baseline jointly optimizes part segmentation along with part transformation parameters, and it does not use $X_{\ell}$. The results in Table 5 show that the joint optimization strategy performs poorly and the proposed alternating strategy is significantly better.
> 3. **R2, I do not consider this work to be concurrent, as it was already published before your submission:** R2 was published in the CVPR proceedings on 17 June after the NeurIPS submission deadline. The reviewer might be referring to the day that it became public in the arXiV, which is 1 April, 1 month and 21 days before our submission. Such a short duration makes it a concurrent work. The same opinion was shared in the review of Reviewer DWX2. We already discussed the differences to R2 in our response and indicated that this method uses two sets of RGB-D views as input and relies on the accurate depth information to estimate 3D object structure unlike our method that uses only RGB views. They do not provide results with only RGB views. Hence it is not comparable to ours.
> 4. **Both R2 and the works they compared can be used as benchmarks for comparison:** We already compared our method to the prior works (Ditto[R3] and PARIS) that use the same input type and supervision. Unlike ours, Ditto [R3] relies on accurate point cloud input which is significantly easier than performing the same task with multi-view input. We already cited and discussed [R3] in our submission.
>
> [R3] Jiang, Zhenyu, Cheng-Chun Hsu, and Yuke Zhu. "Ditto: Building digital twins of articulated objects from interaction." Proceedings of the IEEE/CVF Conference on Computer Vision and Pattern Recognition. 2022.

---

### Official Review · Reviewer_1D4A · 2024-07-08

**Soundness:** 3
**Presentation:** 3
**Contribution:** 3
**Rating:** 6
**Confidence:** 3

**Summary:**

This paper introduces a novel method for decomposing articulated objects and predicting their articulation. The pipeline is trained without supervision. Initially, a "static NeRF" of the object's initial state is obtained. The method then employs part-aware rendering to optimize the pose-change tensor and object segmentation in a decoupled manner. Finally, it reconstructs the voxel grid and performs refinement to achieve high-quality results.

**Strengths:**

1. The pipeline is end-to-end and the training is completely unsupervised.
2. The video results are provided and look impressive.

**Weaknesses:**

1. There are still many artifacts that can be seen in the visualization results
2. The method only works in limited cases. Most of the examples have one or two joints and a large static part.

**Questions:**

1. To optimize the pose-changing tensor, it requires the difference between the target and initial views. What if the object does not have static parts? For example, when scissors cut something, both parts are moving.
2. The results presented in PARIS are not as poor as those in this paper, which the authors should address.

**Limitations:**

The paper discusses the limitations and shows some failure examples.

---

> ### Author Rebuttal · Authors · 2024-08-07
>
> Thanks for reviewing our paper.
>
> 1. **Artifacts for visualization:** ```There are still many artifacts that can be seen in the visualization results```
>
> The artifacts during rendering are indeed caused by the imperfect segmentation in the NeRF space, though they do not harm the pose and joint estimation performance. A challenge of segmenting in the implicit 3D NeRF space is that it is non-trivial to apply regularization to smooth the segmentation results as in standard segmentation methods that work in explicit 2D space. In principle, the artifacts can be reduced with more sets of images from different articulated poses. In future work, we plan to move to explicit 3D representations such as Gaussian Splatting to better regulate the segmentation masks.
>
>
>
> 2. **More evaluation:** ```“the method only works in limited cases. Most of the examples have one or two joints and a large static part.”```
>
> A key advantage of our method over the prior work is its ability to model multiple parts. For the rebuttal, we have further evaluated our model on two more objects, a door with its frame (door-8867) where the frame is static and significantly smaller than the door, from the Sapiens dataset. And the other object is a simple object with 4 chained parts. In the first experiment, our model obtains better performance in all measured metrics compared to PARIS. In the second experiment, we demonstrate our method can model motions of movable parts in chained objects. Yet, how to recover the chained kinematics would be an open question and is out of scope in this work. Detailed results can be found in the attached 1-page PDF. We will include these results along with more such objects in the final version.
>
>
> 3. **Limitation:** ```“What if the object does not have static parts? For example, when scissors cut something, both parts are moving.”```
>
> Indeed our method assumes one part of the object is static to the canonical frame in different articulated poses, which is a common limitation to ours and as well as the prior work. In such cases, the challenge is to segment and track object parts simultaneously. A potential solution could be ingesting a video input with multiple frames and using optical flow to track and cluster parts. However, modeling from video data is out of scope in this paper.
>
> 4. **Baseline performance:** ```The results for PARIS are far worse than the original paper, need to check on it.```
>
> As we pointed out in the footnote in page 6, similar problems for reproducibility for PARIS have been acknowledged as challenging by other research in issues on the official repo. The author’s response to the reproducibility issue in the official code repo is:`
>
> ```“the training for each object is essentially an optimization process, which can be easily affected by the randomness. I was also encountering the unstable training issue you mentioned in my practice”```
>
> For a fair comparison, we follow the default setting provided from the official repo, and reproduce the result on the officially released data. To account for the randomness, we report mean and std metric for 5 independent runs in our paper. In this setting, we can see that our methods are more robust across different objects and delivered more consistent performance compared to PARIS.
>
> Besides, concurrent work [R1] also reports the reproduced performance of PARIS using the officially released data, which also shows similarly lower performance for PARIS.
>
> [R1] Neural Implicit Representation for Building Digital Twins of Unknown Articulated Objects, CVPR 2024

---

> > ### Comment · Reviewer_1D4A · 2024-08-12
> >
> > Thanks for the authors' response. I do not have further concerns and I would like to increase my rating.

---

> > > ### Author Response · Authors · 2024-08-12
> > > **Thanks for the reply**
> > >
> > > Thank you again for your review!

---

### Official Review · Reviewer_DWX2 · 2024-07-09

**Soundness:** 3
**Presentation:** 3
**Contribution:** 3
**Rating:** 6
**Confidence:** 3

**Summary:**

This paper presents an unsupervised framework that jointly learns articulations and part segmentations of objects with rigid parts from multi-view images.

Specifically, they proposed a two-stage approach. In the first stage, a static NeRF is fitted to one of the object states. In the second stage, the optimization alternates between optimizing part assignment and part pose estimations. The key to their approach is a 3D voxel grid heuristic that helps initialize the part assignment.

The proposed method shows promising quantitative and qualitative results in view synthesis, pose estimation, and part segmentation compared to the previous method PARIS. Notably, the proposed method also shows more consistent results across multiple runs.

**Strengths:**

The presented approach makes sense and shows remarkable results compared to the baseline. In particular:
- No labels are required for training.
- A single NeRF plus a small segmentation head is trained, making it more parameter-efficient compared to the baseline (PARIS).
- The learned articulated structures show better geometry accuracy/consistency during motion, compared to the baseline approach.

Overall, the proposed method is promising, the writing is easy to follow, and the problem it tackles is of great importance for a wide range of applications in robotics.

**Weaknesses:**

While the approach looks promising, it does have some room for improvement
- As can be observed in the supplementary video, the segmentation is still noisy, containing floaters that have nothing to do with the moving part.
- Using pixel difference as the heuristic for tagging moving parts is potentially problematic. For example, in a real-world setting, the constantly changing environment light produces color differences across views/states, which can greatly affect the tagging accuracy.
- The examples presented still have a rather simple articulated structure (i.e., no complicated kinematic chains that connect multiple parts). Note that this is an open problem/common issue not specific to the proposed method.
- The approach is limited to rigid deformation (as discussed in L303-304). Learning a deformation field may alleviate this problem, but this is out of the scope of this work.

**Questions:**

Below are some questions that I have:
- A concurrent work [1] tackles the same topic and similarly achieves impressive results. It would be nice if the authors briefly discuss the work (what’s in common, and what set the two apart). Note that [1] was not yet published at the time of NeuRIPS submission, so this is just a nice-to-have discussion, and will not affect the final rating.

Typo:
- L212: geodesic distance should be e_g?

[1] Neural Implicit Representation for Building Digital Twins of Unknown Articulated Objects, CVPR 2024

**Limitations:**

The paper adequately addresses the limitations in Section 5.5 and Section 6. It is also pretty awesome that the paper also presents failure cases and further analysis in Supplementary (A.3)

---

> ### Author Rebuttal · Authors · 2024-08-07
>
> Thanks for reviewing our paper!
>
> 1. **Problematic heuristic**: ```Using pixel difference as the heuristic for tagging moving parts is potentially problematic```
>
> We do not use RGB pixel values but tag the moving parts based on the opacity difference (the alpha channel for RGBA images), which won’t be affected by the RGB values (see line 154). Thus, it works robustly even in adverse and fast-changing lighting conditions. Furthermore, as we shown in the supplementary, Fig 8.a, our pipeline is robust to noisy tagging initialization.
>
> 2. **Simple examples**: ```The examples presented still have a rather simple articulated structure. Note that this is an open problem/common issue not specific to the proposed method.```
>
> Thanks for pointing out this limitation in our work. In case of multiple connected parts, our method would provide a global transformation of parts. In the presence of known kinematic structure, one can estimate the local transformations for each part. We would like to note that structure estimating is an actively studied problem by itself and out of scope for our paper.
>
> 3. **Limitation:** ```The approach is limited to rigid deformation (as discussed in L303-304). Learning a deformation field may alleviate this problem, but this is out of the scope of this work.```
>
> This is indeed a discussed limitation of our method. Rigid articulated objects are ubiquitous in our daily life and hence modeling such objects is a valuable and challenging problem. We plan to explore different deformation parameterizations including non-rigid ones in our future work.
>
> 4. **Discussion for concurrent work [R1]**
>
> The concurrent work uses two sets of RGBD images for different articulation status as input. Thanks to the depth measurements, they could first reconstruct two 3D mesh models for different articulations and then compute part-level transformation based on 3D point correspondences. Part-segmentation is obtained by grouping points that share similar motions. In contrast, our method does not require measurements from a depth sensor and focuses on a more challenging problem setting. Hence we build a NeRF model based on one articulation status and utilize the 2D information from the second image set for part-segmentation in the NeRF space and part motion estimation.
> [R1] Neural Implicit Representation for Building Digital Twins of Unknown Articulated Objects, CVPR 2024

---

> > ### Comment · Reviewer_DWX2 · 2024-08-11
> > **Thanks for the responses.**
> >
> > Thanks for addressing and clarifying the concerns I have.
> >
> > My apology for misunderstanding the pixel difference computation -- certainly using opacity only could avoid the above-mentioned problem. Just one little comment regarding the pixel-difference thing: it would be great if the pixel-difference part in Figure 2 could be improved. Figure 2 depicts that RGB image and foreground mask are used for pixel difference computation. Perhaps it would be better to explicitly mention/depict opacity here, so it would be more accurate/less ambiguous, and align with the text better.
> >
> > Other than that, all my concerns are addressed or properly discussed.

---

> > > ### Author Response · Authors · 2024-08-12
> > > **Thanks for the reply**
> > >
> > > Thank you for your suggestion! We will improve the Figure 2 in the final version for better clarity.

---

### Official Review · Reviewer_JqAE · 2024-07-17

**Soundness:** 3
**Presentation:** 2
**Contribution:** 3
**Rating:** 6
**Confidence:** 3

**Summary:**

The paper proposes a method for modeling articulated objects with neural radiance fields (NeRF). It employs a stage-wise training schema, first building a NeRF of the object in a reference configuration. Then a segmentation in parts and relative pose-changes are learned in an alternating fashion. A modified rendering equation and a corresponding volume sampling strategy are introduced that take into account the rigid deformations of the object parts.

**Strengths:**

The paper considers unsupervised segmentation in parts by adding a segmentation head to the "static NeRF". The latter is learned by using a number of images capturing the object in a reference configuration, while part segmentation and relative transformations are learned by using a number of images capturing the object in a different configuration.

The proposed method approaches the problem of articulated object reconstruction with stage-wise training, while segmentation and pose estimation are learned in a decoupled way. The experimental evaluation shows that this approach performs better with respect to competing methods. In addition, the proposed method is able to reconstruct objects with multiple parts.

**Weaknesses:**

The text contains a few typos and other errors, like incomplete sentences, missing verbs etc. Some editing of the text is required.
Additionally, the clarity of the presentation can be improved. It is true that the stage-wise learning schema is somewhat involved, hence some additional effort should be devoted in explaining the steps as clear as possible, especially in Section 4.2. One suggestion is to provide a more detailed explanation of Figure 3, and provide dimensions of matrices U, F etc. Also, the formula in L.167-168 does not seem to be correct, especially regarding how the viewpoint v' is used. Should v' be an argument of a function?

Another aspect that is missing regards the efficiency of the proposed method. It is important to discuss training and inference times in comparison to other baselines. On a similar note, a discussion on how the number of iterations used at each stage affects the final result could be included in the ablations.

Regarding experimental evaluation, ideally, comparison could also consider state-of-the-art articulated object reconstruct methods which are not based on NeRFs, such as  [R1] and [R2].

Finally, not enough details are shared with respect to the models used, making reproducibility of the approach challenging.

[R1] Kawana & Harada, Detection based part-level articulated object reconstruction from single RGBD image. NeurIPS 2023

[R2] Kawana, Mukuta, & Harada, Unsupervised pose-aware part decomposition for man-made articulated objects, ECCV 2022

### Minor comments
- L.10: incomplete sentence
- L.71: needs rephrasing
- L.105: "And the opacity value ..." incomplete
- L.199: It is fine that the same subset of 3D PartNet-Mobility dataset used in [15] is considered for the comparison, but it would be interesting to report results on a wider selection of objects
- Section 5: the order in which figures are presented is not following the text, this creates some confusion to the reader
- L.270: "performs consistently performs"
- L.292: "presented in second and third row" does not follow the structure of the figure

**Questions:**

- How does the number of iterations used at each stage affect the final result? Is the method sensitive to these hyper-parameters?
- Can the method handle joints with more than one degree of freedom?

**Limitations:**

Limitations are discussed in the text. I think it would be important to discuss also challenges and limitations of multi-part object reconstruction.

---

> ### Author Rebuttal · Authors · 2024-08-07
>
> Thanks for the review!
>
> 1. **Typos and other errors**: Thanks ! We will fix them in the final version.
>
> 2. **Clarity:** ```explaining the steps in Section 4.2, provide dimensions of matrices U, F, the formula in L.167-168 ... how the viewpoint v' is used```
>
> We use homogeneous coordinates for representing $X_{\ell}$ and convert it to a $4 \times k_\ell$ matrix by appending an extra 1 as scale to each of its 3D columns before plugging into the projection formula $U_{\ell} = K M_{\ell}^{-1} {v'} X_{\ell}$. Let $k_\ell$ denote the number of 3D points in $X_\ell$. In the projection formula, camera intrinsic parameters $K$ is $ 3 \times 4$, the inverse part-pose transformation matrix $M_\ell^{-1}$ in shape $4 \times 4$, camera viewpoint (extrinsic parameter) $\mathbf{v’}$ in shape $4 \times 4$, the projection result $U_\ell$ in shape $3 \times k_\ell$.  Finally, we normalize each column of $U_\ell$ with its last entry to recover the real pixel location after projection. We will clarify these points in the final version.
>
> 3. **Efficiency**: ```Another aspect that is missing regards the efficiency of the proposed method```
>
> Our method takes on average about 15 minutes. The inference time for 2-, 3-, 4-part objects are 0.58, 0.87, 1.23 seconds. While the PARIS takes 5 minutes and 6 VRAM for training and the inference time for a 2-part object is 10.20 seconds on average. The above figures are tested with AMD 7950x CPU and Nvidia RTX 4090 GPU. We will include these details in the supplementary.
>
> 4. **Hyperparameter sensitivity**: ```How does the number of iterations ... Is the method sensitive to these hyper-parameters?```
>
> We use the same number of iterations, 6000 for optimizing $M_\ell$ and 2000 for optimizing $s_\ell$ in our method over all object instances (see Line 193). This ensures that the training converges. We observe that the variations in the results are negligible after the given number of iterations. We will provide a quantitative sensitivity analysis in the final version.
>
> 5. **Implementation details**: ```not enough details with respect to the models```
>
> We built our part-aware NeRF model based on the proposal estimator in [R3] and the NGP-based radiance field in [R4]. We extended the resulting model for segmentation by adding 2-layer MLPs with ReLU activation and the hidden dimensionality 64 in both estimator and radiance field. More details will be added to the supplementary. We will further release our code and data with the final version.
>
> [R3] Mip-NeRF 360: Unbounded Anti-Aliased Neural Radiance Fields. CVPR 2022
>
> [R4] Instant Neural Graphics Primitives. ACM Trans. Graph. 2022
>
> 6. **Comparison to prior work**: ```compared to state-of-the-art articulated object reconstruct methods ... such as [R1] and [R2]```
>
> Thanks for pointing out the related work. The method in [R1] learns to model articulated objects from a single RGBD image by detecting parts and reconstructing parts separately. Unlike our method that is unsupervised, it is supervised and requires ground-truth for joint parameters, joint-type and part-level oriented bounding boxes. In addition, our method estimates 3D from a set of 2D images without requiring measurements from a depth sensor.  The method in [R2] is an unsupervised method that learns to model articulated objects from point cloud data. It assumes a dense and accurate point cloud per object instance as input, while our method focuses on the more challenging task of estimating 3D from a set of 2D images alone. Hence, their results are not comparable to ours due to different input types and levels of supervision. We will include these works in our related work section.
>
> [R1] Kawana & Harada, Detection based part-level articulated object reconstruction from single RGBD image. NeurIPS 2023
>
> [R2] Kawana, Mukuta, & Harada, Unsupervised pose-aware part decomposition for man-made articulated objects, ECCV 2022
>
> 7. **More extensive experiments**: ```... report results on a wider selection of objects```
>
> Thanks for the suggestion. We reported our results in the benchmark that was used by the prior work, and additionally evaluated our method on objects with multiple parts and a real-object in the submission. We also provided an additional experiment on doors where the static part is smaller and a simple 4-parts chained object in the one-page pdf. In the final version, we will include examples from more categories such as door, fan, monitor that have not been included in the original subset.
>
> 8. **Modelling multiple degrees of freedom**: ```Can the method handle joints with more than one degree of freedom?```
>
> Our method is not limited to estimating a single degree of freedom (DoF), as we do not restrict the part transformations to a single DoF. However, the dataset included single DoF joint objects only. Hence, we provide an extra experiment of a chained 4-parts object and estimate the 6 DoF $se(3)$ transformation for each movable part in the one-page PDF.
>
>
> 9. **Limitations:** ```... to discuss also challenges and limitations of multi-part object reconstruction.```
>
> One challenge for multi-part object reconstruction is to reconstruct the occluded parts, as occlusions between parts are more likely to occur more when the number of parts increases. For example, in the two-drawer storage shown in Figure 11 of the supplementary, the drawers will be partially occluded in both articulation status.
> Another challenge would be about taking physical constraints into consideration. For example, the segmented parts should not have collisions with other parts, or the connected joints should not be detached during articulation pose estimation. Third, recovering complex kinematic structures like chained parts is still an open problem. Finally, a dataset consisting of objects with more complex kinematic structures is currently missing. Such a dataset would enable advancing the progress in articulation object modeling.

---

> ### Comment · Reviewer_JqAE · 2024-08-13
> **Post-rebuttal comments**
>
> The author responses have addressed my main concerns and clarified some important aspects regarding notation, comparison to prior work and implementation details. For these reasons, I increase my rating to 6.

---

### Author Rebuttal · Authors · 2024-08-07

# Global comment
We sincerely thank all reviewers for their valuable and insightful comments. We are particularly encouraged by the positive feedback received and appreciate the opportunity to address the concerns raised. We are happy to address these common issues comprehensively in the following sections.
## More examples for evaluation (JqAE, DWX2, 1D4A).
- Reviewer **JqAE**: ```it would be interesting to report results on a wider selection of objects.```
- Reviewer **DWX2**: ```The examples presented still have a rather simple articulated structure.```
- Reviewer **1D4A**: ```the method only works in limited cases. Most of the examples have one or two joints and a large static part.```

To address the concerns for evaluation, we carried out extra experiments for the object door with its frame, which is not included in the PARIS evaluation set (**JqAE**) and it has a larger movable part compared to the static part(**1D4A**). The quantitative results of our methods consistently outperform the PARIS.
Additionally, we use our method to estimate the motions of 4-part chained objects. Since recovering the chained kinematic structure is out of scope in our work, we estimate the 6-DoF se(3) transform for each movable part (**JqAE**, **DWX2**, **1D4A**). Qualitative evaluations show the robustness of motion estimation of our method and details can be found in the one-page PDF.
## Artifacts for visualization (DWX2, 1D4A)
- Reviewer **DWX2**: ```As can be observed in the supplementary video, the segmentation is still noisy, containing floaters that have nothing to do with the moving part.```
- Reviewer **1D4A**: ```There are still many artifacts that can be seen in the visualization results```

The artefacts during rendering are indeed caused by the imperfect segmentation in the NeRF space, though they do not harm the pose and joint estimation performance. A challenge of segmenting in the implicit 3D NeRF space is that it is non-trivial to apply regularization to smooth the segmentation results as in standard segmentation methods that work in explicit 2D space. In principle, the artefacts can be reduced with more images from different articulated poses. In future work, we plan to move to explicit 3D representations such as Gaussian Splatting to regulate the segmentation masks better.

---

### Decision · Program_Chairs · 2024-09-25

**Decision:**

Accept (poster)

**Comment:**

Post rebuttal, reviews were split. Three reviewers were in favor of accepting the paper. They argued the work tackled an important and challenging problem, used a method that would be a good contribution, thought the experiments demonstrated the value of the work, and overall believed that the paper would be a good addition to the overall literature.

One reviewer was strongly opposed to the work, claiming that the method wasn't a substantial enough contribution to the literature, and that it should be compared with more work. One concern was the paper's baseline results differed from the published results. The AC is convinced by the authors' explanation of that difference is by the instability of the baseline methods (which was noted in the NeurIPS submission, in the github issues for the paper in question, and acknowledged in other papers too). The authors' approach of running methods five times and reporting the mean and standard deviation seems a sensible mitigation. A second point of disagreement was a comparisons with a CVPR 24 paper that appeared on Arxiv before the NeurIPS submission deadline. The AC agrees with the authors and one of the other reviewers that the paper is concurrent, following the NeurIPS author FAQ's guideline of 2 months. Comparisons might be a nice addition for the paper's impact, but are not necessary.

Overall, the AC agrees with the reviewers recommending acceptance. This decision was based on the paper, the reviews, the rebuttal, and the discussion after the rebuttal. The AC asks the authors to incorporate their responses to the reviewers into the final version of the paper.